# Modelling biological assays with Adaptive Deep Kernel Learning

## Abstract

Due to the significant costs of data generation, many prediction tasks within drug discovery are by nature few-shot regression (FSR) problems, including accurate modelling of biological assays. Although a number of few-shot classification and reinforcement learning methods exist for similar applications, we find relatively few FSR methods meeting the performance standards required for such tasks under real-world constraints. Inspired by deep kernel learning, we develop a novel FSR algorithm that is better suited to these settings. Our algorithm consists of learning a deep network in combination with a kernel function and a differentiable kernel algorithm. As the choice of kernel is critical, our algorithm learns to find the appropriate kernel for each task during inference. It thus performs more effectively with complex task distributions, outperforming current state-of-the-art algorithms on both toy and novel, real-world benchmarks that we introduce herein. By introducing novel benchmarks derived from biological assays, we hope that the community will progress towards the development of FSR algorithms suitable for use in noisy and uncertain environments such as drug discovery.

## 1    Introduction

Following breakthroughs in domains including computer vision, autonomous driving, and natural language processing, deep learning methods are now entering the domain of pharmaceutical R&D. Recent successes include the deconvolution of biological targets from -omics data (Min et al., 2017), generation of drug-like compounds via *de novo* molecular design (Xu et al., 2019), chemical synthesis planning (Segler and Waller, 2017; Segler et al., 2017), and multi-modal image analysis for quantification of cellular response (Min et al., 2017). A common characteristic of these applications, however, is the availability of high quality, high quantity training data. Unfortunately, many critical prediction tasks in the drug discovery pipeline fail to satisfy these requirements, in part due to resource and cost constraints (Cherkasov et al., 2014).

We therefore focus this work on modelling biological assays (bio-assays) relevant in the early stages of drug discovery, primarily binding and cellular readouts. Under the constraints of an active drug discovery program, the data from these assays, consisting of libraries of molecules and their associated real-valued activity scores, is often relatively small and noisy (refer to statistics in Section 5). In many contexts, it can be challenging to build a training set of even a few dozen samples per individual assay. Modelling an assay is thus best viewed as a few-shot regression (FSR) problem, with many variables (including experimental conditions, readouts, concentrations, and instrument configurations) accounting for the data distribution being generated. Practically, these variables make it infeasible to compare data collected across different assays, thereby making it difficult to learn predictive models from molecular structures. Furthermore, as bio-assay modelling is intended to be used for prioritizing molecules for subsequent evaluation (e.g. Bayesian optimization) and efficiently exploring the overall chemical space (e.g. active learning), accurate prediction and uncertainty estimation using few datapoints is critical to successful application in drug discovery.

It is our view that robust FSR algorithms are needed to tackle this challenge. Specifically, we argue that these algorithms should remain accurate in noisy environments, and also provide well-calibrated uncertainty estimates to inform efficient exploration of chemical space during molecular optimization. Fortunately, recent advances in few-shot learning have led to new algorithms that learn efficiently and generalize adequately from small training data (Wang and Yao, 2019; Chen et al., 2019). Most

have adopted the meta-learning paradigm (Thrun and Pratt, 1998; Vilalta and Drissi, 2002), where some prior knowledge is learned across a large collection of tasks and then transferred to new tasks in which there are limited amounts of data. Such algorithms tend to differ in two aspects: the **nature of the meta-knowledge captured** and the **amount of adaptation performed at test-time** for new tasks or datasets. The meta-knowledge refers to the domain specific prior needed to solve each task most effectively. Due to the size of the total chemical space accessible when modelling bio-assays (Bohacek et al., 1996), there is a particular need for the meta-knowledge to be sufficiently rich so as to allow for extrapolation and uncertainty estimation in unseen regions of chemical space at test-time (i.e. for new tasks). Given that the same molecule can behave differently across different assays, greater test-time adaptation is also required and must be accounted for during modelling.

In previous work, metric learning methods (Koch et al., 2015; Vinyals et al., 2016; Snell et al., 2017; Garcia and Bruna, 2017; Bertinetto et al., 2018) accumulate meta-knowledge in high capacity covariance/distance functions and use simple base-learners such as k-nearest neighbor (Snell et al., 2017; Vinyals et al., 2016) or low capacity neural networks (Garcia and Bruna, 2017) to produce adequate models for new tasks. However, they do not adapt the covariance functions nor the base-learners at test-time. Initialization- and optimization-based methods (Finn et al., 2017; Kim et al., 2018; Ravi and Larochelle, 2016) that learn the initialization points and update rules for gradient descent-based algorithms, respectively, allow for improved adaptation on new tasks but remain time consuming and memory inefficient. We therefore argue that to ensure optimal performance when modelling bio-assays, it is crucial to combine the strengths of both types of methods while also allowing for the incorporation of domain-specific knowledge when making predictions. We achieve this by framing FSR as a deep kernel learning (DKL) task, deriving novel algorithms that we apply to modelling specific assays and readouts.

**Contributions:** Our contributions are three-fold. We first frame few-shot regression as a DKL problem and showcase its advantages relative to classical metric learning methods. We then derive the adaptive deep kernel learning (ADKL) framework by learning a conditional kernel function that is task dependant, allowing for more test-time adaptation than the DKL framework. Finally, we introduce two real-world datasets for modelling biological assays using FSR. With this contribution, we hope to encourage the development of subsequent few-shot regression methods suitable for real-world applications, as is the case for few-shot classification and reinforcement learning, each of which have received comparatively greater attention in recent years (Wang and Yao, 2019).

## 2 DEEP KERNEL LEARNING

In this section, we describe the DKL framework introduced for single tasks by Wilson et al. (2016). We then extend it to few-shot learning and discuss its advantages over metric learning algorithms.

**Single Task DKL:** Let $D_{trn}^t = \{(\mathbf{x}_i, y_i)\}_{i=1}^m \subset \mathcal{X} \times \mathbb{R}$, a training dataset available for learning the regression task $t$ where $\mathcal{X}$ is the input space and $\mathbb{R}$ is the output space. A DKL algorithm aims to obtain a non-linear embedding of inputs in the embedding space $\mathcal{H}$, using a deep neural network $\phi_{\boldsymbol{\theta}} : \mathcal{X} \to \mathcal{H}$ of parameters $\boldsymbol{\theta}$. It then finds the minimal norm regressor $h_*^t$ in the reproducing kernel Hilbert space (RKHS) $\mathcal{R}$ on $\mathcal{H}$, that minimize an objective function such as

$$h_*^t := \operatorname*{argmin}_{h \in \mathcal{R}} \lambda \|h\|_{\mathcal{R}} + \ell(h, D_{trn}^t) \tag{1}$$

where $\ell$ is a non-negative loss function that measures the loss of a regressor $h$ and $\lambda$ weighs the importance of the norm minimization against the training loss. Following the representer theorem (Scholkopf and Smola, 2001; Steinwart and Christmann, 2008), $h_*^t$ can be written as a finite linear combination of kernel evaluations on training inputs, i.e.:

$$h_*^t(\mathbf{x}) = \sum_{(\mathbf{x}_i, y_i) \in D_{trn}^t} \alpha_i^t k_{\boldsymbol{\rho}}(\phi_{\boldsymbol{\theta}}(\mathbf{x}), \phi_{\boldsymbol{\theta}}(\mathbf{x}_i)), \tag{2}$$

where $\boldsymbol{\alpha}^t = (\alpha_1^t, \cdots, \alpha_m^t)$ are the learned combination weights and $k_{\boldsymbol{\rho}} \colon \mathcal{H} \times \mathcal{H} \to \mathbb{R}_+$ is a chosen reproducing kernel of $\mathcal{R}$ with hyperparameters $\boldsymbol{\rho}$. Candidate kernels include the radial basis, polynomial, and linear kernels. $\boldsymbol{\alpha}^t$ can be obtained by using a differentiable kernel method enabling the computation of the gradients of the loss w.r.t. the parameters $\boldsymbol{\theta}$. Such methods include Gaussian Process (GP), Kernel Ridge Regression (KRR), and Logistic Regression (LR).

As DKL inherits from deep learning and kernel methods, it follows that gradient descent algorithms are required to optimize the network parameters $\boldsymbol{\theta}$. The latter can be high dimensional and a substantial amount of training samples are required to train DKL models and avoid overfitting. However, once the latter condition is met, scalability of the kernel method can be limiting (running time in $O(m^3)$ for $m$ training samples) and approximations can be needed for scalability (see Williams and Seeger (2001); Wilson and Nickisch (2015)).

**Few-Shot DKL:** In few-shot learning, one has access to a meta-training collection $\mathscr{D}_{meta-trn} :=$ $\left\{ (D_{trn}^{t_j}, D_{val}^{t_j}) \right\}_{j=1}^{T}$ of $T$ tasks Each task $t_j$ has its own training (or support) set $D_{trn}^{t_j}$ and validation (or query) set $D_{val}^{t_j}$. A meta-testing collection $\mathscr{D}_{meta-tst}$ is also available to assess the generalization performance of the few-shot algorithm across unseen tasks. To obtain a Few-Shot DKL (FSDKL) method for FSR in such settings, one can share the parameters of $\boldsymbol{\phi_\theta}$ across all tasks, similar to metric learning algorithms. Hence, for a given task $t_j$, the inputs are first transformed by the function $\boldsymbol{\phi_\theta}$ and then a kernel method is used to obtain the regressor $h_*^{t_j}$, which will be evaluated on $D_{val}^{t_j}$. Here, KRR and GP are explored as they are the state-of-the-art algorithms for kernel-based regression. The latter is used to allow our models to provide accurate predictive uncertainty, which is useful when prioritizing molecules in the context of drug discovery.

**KRR:** Using the squared loss and the L2-norm to compute $\|h\|_{\mathcal{R}}$, KRR gives the optimal regressor for a task $t$ and its validation loss $\mathcal{L}_{\boldsymbol{\theta},\boldsymbol{\rho},\lambda}^{t}$ as follows:

$$h_*^t(\mathbf{x}) = \boldsymbol{\alpha} K_{\mathbf{x},trn}, \quad with \quad \boldsymbol{\alpha} = (K_{trn,trn} + \lambda I)^{-1} \mathbf{y}_{trn} \tag{3}$$

$$\mathcal{L}_{\boldsymbol{\theta},\boldsymbol{\rho},\lambda}^{t} = \mathop{\mathbf{E}}_{\mathbf{x},y \sim D_{val}^t} (\boldsymbol{\alpha} K_{\mathbf{x},trn} - y)^2, \tag{4}$$

where $\mathbf{y}_{trn} = (y_1, \cdots, y_{|D_{trn}^t|})^T$, $K_{trn,trn}$ is the matrix of kernel evaluations where entry $i, l$ is $k_{\boldsymbol{\rho}}(\boldsymbol{\phi_\theta}(\mathbf{x}_i), \boldsymbol{\phi_\theta}(\mathbf{x}_l))$ for pairs of examples in $D_{trn}^t$. An equivalent definition applies to $K_{\mathbf{x},trn}$.

**GP:** When using the negative log likelihood loss function, the GP algorithm gives a probabilistic regressor for which the predictive mean, covariance, and loss for a task $t$ are:

$$\mathcal{L}_{\boldsymbol{\theta},\boldsymbol{\rho},\lambda}^{t} = -\ln \mathcal{N}(\mathbf{y}_{val}; \mathbb{E}[h_*^t], \mathrm{cov}(h_*^t)), \tag{5}$$

$$\mathbb{E}[h_*^t] = K_{val,trn}(K_{trn,trn} + \lambda I)^{-1}\mathbf{y}_{trn}, \tag{6}$$

$$cov(h_*^t) = K_{val,val} - K_{val,trn}(K_{trn,trn} + \lambda I)^{-1}K_{trn,val} \tag{7}$$

Finally, the parameters $\boldsymbol{\theta}$ of the neural network, along with $\lambda$ and the kernel hyperparameters $\boldsymbol{\rho}$, are optimized using the expected loss on all tasks:

$$\mathop{\mathrm{argmin}}_{\boldsymbol{\theta},\boldsymbol{\rho},\lambda} \mathop{\mathbf{E}}_{t \sim \mathscr{D}_{meta-trn}} \mathcal{L}_{\boldsymbol{\theta},\boldsymbol{\rho},\lambda}^{t}. \tag{8}$$

To summarize, FSDKL finds a representation common to all tasks such that the kernel method (in our case, GP and KRR) will generalize well from a small amount of samples. In doing so, this alleviates two of the main limitations of single task DKL: i) the scalability of the kernel method is no longer an issue since we are in the few-shot learning regime[1], and ii) the parameters $\boldsymbol{\theta}$ (and $\boldsymbol{\rho}$, $\lambda$) are learned across a potentially large amount of tasks and samples, providing the opportunity to learn a rich representation without overfitting.

Despite shared characteristics with the metric learning framework, the FSDKL framework is more powerful and flexible. It provides better task-specific adaptation due to the inference of the appropriate model using the kernel methods compared to shared model parameters in metric learning. After meta-training, any task-specific model also inherits the generalization guarantees of kernel-based models, and consequently increasing the number of shots for new tasks can only improve generalization performance. The incorporation of prior knowledge through user-specific kernel functions is also a major advantage of DKL over metric learning (e.g. use periodic kernels for periodic function regression tasks).

---

[1]Even with several hundred samples, the computational cost of embedding each example is usually higher than inverting the Gram matrix.

126 ## 3 ADAPTIVE DEEP KERNEL LEARNING

127 In this section, we present a new algorithm, deemed adapative deep kernel learning (ADKL). Funda-
128 mentally, it differs from FSDKL by having more flexibility in its kernel definition and by learning to
129 produce task-specific kernel functions during the meta-training instead of using one defined by the
130 user. It does so by learning a task representation using a task encoding network $\psi_{\boldsymbol{\eta}}$ and leveraging it
131 to build task-specific kernels using a multi-modal neural network $c_{\boldsymbol{\rho}}$. More explicitly, given a task $t$,
132 ADKL first computes a task embedding $\mathbf{z}_t = \psi_{\boldsymbol{\eta}}(D_{trn}^t)$ using its support set $D_{trn}^t$ and then it infers
133 the adapted kernel with $c_{\boldsymbol{\rho}}$. We describe in more detail both the task encoding network $\psi_{\boldsymbol{\eta}}$ and the
134 network $c_{\boldsymbol{\rho}}$ responsible for computing the task-specific kernel below.

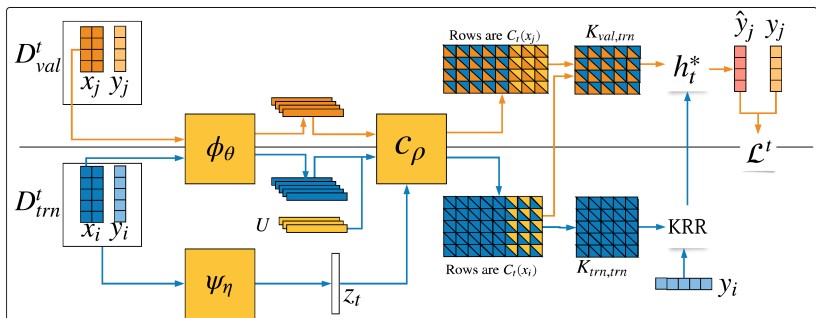

Figure 1: ADKL-KRR. The blue and orange colors show the procedure for a task during internal train
and test, respectively. During training, ADKL first computes a task embedding $\mathbf{z}_t = \psi_{\boldsymbol{\eta}}(D_{trn}^t)$ that is
used with a pseudo-representations $U$ by the network $c_{\boldsymbol{\rho}}$ to produce a the task-specific kernel function.
The empirical kernel map of this kernel gives the function $C_t(\cdot)$ that is evaluated for every training point
to produce $K_{trn,trn}$. The latter and the train targets are used by KRR (or GP) to produce the model $h_t^*$.
At evaluation, $C_t(\cdot)$ is evaluated again for every test point to obtain $K_{val,trn}$, which is used to compute
the predictions. The loss is then computed and used to update all parameters of ADKL.

135 ### 3.1 TASK ENCODING

136 The challenge of the network $\psi_{\boldsymbol{\eta}}$ is to capture complex dependencies in the training set $D_{trn}^t$ to
137 provide a useful task encoding $\mathbf{z}_t$. Furthermore, the task encoder should be invariant to permutations
138 of the training set and be able to encode a variable amount of samples. After exploring a variety
139 of architectures, we found that those that are more complex, such as Transformers (Vaswani et al.,
140 2017), tend to underperform. This is possibly due to overfitting or the sensitivity of training such
141 architectures.

142 Consequently, inspired by DeepSets (Zaheer et al., 2017), we propose a simple order invariant
143 network that captures the first and second order statistics of regression datasets. Given a dataset, this
144 network first processes each of its samples individually as follows: a) extract input features using
145 $\phi_{\boldsymbol{\theta}}$ (see section 2), b) concatenate the input features with the target and embed the obtained vector
146 using a simple fully connected neural network $\mathbf{r}_{\boldsymbol{\eta}}$ of parameters $\boldsymbol{\eta}$ [2]. It then computes the first and
147 the second order statistics of the obtained vectors for all samples of the dataset and concatenates them
148 to produce the representation. More formally,

$$\psi_{\boldsymbol{\eta}}(D_{trn}^t) := \left[\mu^t, \sigma^t\right], \ with \ \mu^t = \mathop{\mathbf{E}}_{(\mathbf{x},y)\in D_{trn}^t} \mathbf{r}_{\boldsymbol{\eta}}([\phi_{\boldsymbol{\theta}}(\mathbf{x}_i), y_i]), \quad \sigma^t = \mathop{\mathbf{Var}}_{(\mathbf{x},y)\in D_{trn}^t} \mathbf{r}_{\boldsymbol{\eta}}([\phi_{\boldsymbol{\theta}}(\mathbf{x}_i), y_i])$$

149 where $[\cdot, \cdot]$ is the concatenation operator. As $\mu^t$ and $\sigma^t$ are invariant to permutations in $D_{trn}^t$, it
150 follows that $\psi_{\boldsymbol{\eta}}$ is also permutation invariant. Overall, $\psi_{\boldsymbol{\eta}}$ is simply the concatenation of the first and
151 second moments of the sample representations, which were nonlinear transformations of the original
152 inputs and targets.

To help the training of the parameters $\boldsymbol{\eta}$, we add a regularization term that maximizes the mutual
information between $D_{trn}^t$ and $D_{val}^t$. This encourage the network to produce similar task encodings

---

[2]These are the only parameters involved in the computation of the task encoding, which is why we also use
the notation $\psi_{\boldsymbol{\eta}}$.

when presented with different data partitions for a given task. Concretely, we maximize the lower bound on the mutual information between the task representations given by the support and the query sets instead of the true mutual information (Belghazi et al., 2018). Using a batch of $b$ tasks and the cosine similarity $c$ as the similarity measure the between two task encodings, this lower bound $\tilde{I}_{\boldsymbol{\eta}}$ is defined by Eq. (9) and is the regularizer that we used to a have better task encoder.

$$\tilde{I}_{\boldsymbol{\eta}} \overset{\text{def}}{=} \frac{1}{b} \sum_{j=1}^{b} c(\boldsymbol{\psi}_{\boldsymbol{\eta}}(D_{trn}^{t_j}), \boldsymbol{\psi}_{\boldsymbol{\eta}}(D_{val}^{t_j})) - \ln \frac{1}{b(b-1)} \sum_{j=1}^{b} \sum_{i \neq j} e^{c(\boldsymbol{\psi}_{\boldsymbol{\eta}}(D_{trn}^{t_j}), \boldsymbol{\psi}_{\boldsymbol{\eta}}(D_{val}^{t_i}))} \tag{9}$$

## 3.2 TASK-SPECIFIC KERNEL

Here, we describe how the task-specific kernels are inferred using the task representations described previously. In fact, they are all obtained using a multi-modal neural network $c_{\boldsymbol{\rho}}$ of parameter $\boldsymbol{\rho}$. Given any pair of input representations ($\boldsymbol{\phi}(\mathbf{x})$ and $\boldsymbol{\phi}(\mathbf{x}')$) and a task encoding $\mathbf{z}_t$, this network simply computes the input pair similarity under the condition given by the task encoding as follows:

$$c_{\boldsymbol{\rho}}(\boldsymbol{\phi}(\mathbf{x}), \boldsymbol{\phi}(\mathbf{x}'), \mathbf{z}_t) := MLP_{\boldsymbol{\rho}}([(\boldsymbol{\phi}(\mathbf{x}) - \boldsymbol{\phi}(\mathbf{x}'))^2, \mathbf{z}_t]), \tag{10}$$

where $(\boldsymbol{\phi}(\mathbf{x}) - \boldsymbol{\phi}(\mathbf{x}'))^2$ is the element-wise L2 distance between the input representations, $[\cdot, \cdot]$ is the concatenation operator and $MLP$ is a fully connected neural network of parameters $\boldsymbol{\rho}$ which has a single neuron at its last layer. It bears mentioning that $c_{\boldsymbol{\rho}}$ is symmetric and stationary with regard to $\boldsymbol{\phi}(\mathbf{x})$ and $\boldsymbol{\phi}(\mathbf{x}')$ as their element-wise L2 distances vector is received as part of the input of the fully connected network. Further, by simply concatenating the task representation $\mathbf{z}_t$ to this distance vector at the input, $c_{\boldsymbol{\rho}}$ provides a powerful approach to produce task-specific kernels. However, these kernels are not positive semi-definite (PSD) and cannot be directly used for KRR and GP. Therefore, using the empirical kernel mapping technique (Schölkopf et al., 1999) we computed the task-specific PSD kernel $k_{\boldsymbol{\rho},t}$ associated with a given task representation $\mathbf{z}_t$ obtained from $D_{trn}^t$. This kernel can be written as the empirical kernel map of $c_{\boldsymbol{\rho}}(\cdot, \cdot, \mathbf{z}_t)$ with regard to $D_{trn}^t$ i.e.:

$$k_{\boldsymbol{\rho},t}(\mathbf{x}, \mathbf{x}') = C_t(\mathbf{x}) \cdot C_t(\mathbf{x}'), \quad with$$
$$C_t(\mathbf{x}) = (c_{\boldsymbol{\rho}}(\mathbf{x}, \mathbf{x}_1, \mathbf{z}_t), \cdots, c_{\boldsymbol{\rho}}(\mathbf{x}, \mathbf{x}_m, \mathbf{z}_t)), \quad and \quad (\mathbf{x}_i, \cdot) \in D_{trn}^t \forall i = 1, \cdots, m \tag{11}$$

Using the empirical kernel map of $c_{\boldsymbol{\rho}}$ to compute $k_{\boldsymbol{\rho},t}$ offers the opportunity to introduce *pseudo-input representations* (or *pseudo-representations*) that could improve the kernel evaluations, specially in low data settings. More precisely, instead of computing the empirical kernel map with regard to $D_{trn}^t$ alone, we use $(D_{trn}^t \cup U)$ where $U$ is the set of pseudo-representations . The function $C_t$, from Eq. (11), becomes:

$$C_t(\mathbf{x}) = (c_{\boldsymbol{\rho}}(\mathbf{x}, \mathbf{x}_1, \mathbf{z}_t), \cdots, c_{\boldsymbol{\rho}}(\mathbf{x}, \mathbf{x}_m, \mathbf{z}_t), c_{\boldsymbol{\rho}}(\mathbf{x}, \mathbf{u}_1, \mathbf{z}_t), \cdots, c_{\boldsymbol{\rho}}(\mathbf{x}, \mathbf{u}_l, \mathbf{z}_t)),$$
$$with \quad \mathbf{u}_l \in U \, \forall l = 1, \cdots, |U| \quad and (\mathbf{x}_i, \cdot) \in D_{trn}^t \forall i = 1, \cdots, m \tag{12}$$

The number of pseudo-representations is a hyperparameter of ADKL (in our experiments we choose $|U| \in [0, 50]$) and all pseudo-representations $\mathbf{u}_l \in \mathcal{H}$ are learnable parameters that are shared by all tasks and learned during meta-training. To prevent their collapse into a single point during the training and to ensure that they are well distributed in the feature space $\mathcal{H}$, we add a regularization term ($\tilde{D}_{\mathbf{u}}$) to the training loss function. To introduce this regularization term, let's consider $p$ and $q$ to be the distributions that generate the true input representations and the pseudo-representations, respectively. We make the assumption that $p$ and $q$ are both multivariate Gaussian distributions with diagonal covariance matrices and have respective parameters ($\mu_{\boldsymbol{\phi}}, \sigma_{\boldsymbol{\phi}}^2$) and ($\mu_{\mathbf{u}}, \sigma_{\mathbf{u}}^2$). The parameters of $p$ are estimated using the running means and variances of all input representations computed over batches of tasks. Those of $q$ are estimated using $U$. As, $p$ and $q$ must be close, the training of the pseudo-representations is regularized by minimizing the KL distance $\tilde{D}_{\mathbf{u}}$ between $p$ and $q$, i.e.:

$$\tilde{D}_{\mathbf{u}} = KL(\mathcal{N}(\mu_{\mathbf{u}}, \sigma_{\mathbf{u}}^2) \| \mathcal{N}(\mu_{\boldsymbol{\phi}}, \sigma_{\boldsymbol{\phi}}^2)) \tag{13}$$

Putting it all together, the ADKL training objective is the following:

$$\underset{\boldsymbol{\theta}, \boldsymbol{\eta}, \boldsymbol{\rho}, \mathbf{u}, \lambda}{\operatorname{argmin}} \underset{t_j \sim B}{\mathbf{E}} \mathcal{L}_{\boldsymbol{\theta}, \boldsymbol{\eta}, \boldsymbol{\rho}, \mathbf{u}, \lambda}^{t_j} - \gamma_{task} \tilde{I}_{\boldsymbol{\eta}} \cdot + \gamma_{pseudo} \tilde{D}_{\mathbf{u}}, \tag{14}$$

with $\gamma_{task} \geq 0$ as a tradeoff hyperparameter for the regularization of the task-encoder and $\gamma_{pseudo} \geq 0$ as a tradeoff hyperparameter for the regularization of the pseudo-inputs.

## 4 RELATED WORK

Across the spectrum of learning approaches, DKL methods lie between neural networks and kernel methods. While neural networks can learn from a very large amount of data without much prior knowledge, kernel methods learn from fewer data when given an appropriate covariance function that accounts for prior knowledge of the relevant task. In the first DKL attempt, Wilson et al. (2016) combined GP with CNN to learn a covariance function adapted to a task from large amounts of data, though the large time and space complexity of kernel methods forced the approximation of the exact kernel using KISS-GP (Wilson and Nickisch, 2015). Dasgupta et al. (2018) have demonstrated that such approximation is not necessary using finite rank kernels. Here, we show that learning from a collection of tasks (FSR mode) does not require any approximation when the covariance function is shared across tasks. This is an important distinction between our study and other existing studies in DKL, which learn their kernel for single task applications instead of multiple task collections.

On the spectrum between NNs and kernel methods we must also mention metric learning. Metric learning algorithms learn an input covariance function shared across tasks but rely only on the expressive power of DNNs. First, stochastic kernels are built out of shared feature extractors and simple pairwise metrics (e.g. cosine similarity (Vinyals et al., 2016), Euclidean distance (Snell et al., 2017)), or parametric functions (e.g. relation modules (Sung et al., 2018), graph neural networks (Garcia and Bruna, 2017; Kim et al., 2019a)). Then, within tasks, the predictions are distance-weighted combinations of the training labels with the stochastic kernel evaluations—no adaptation is done.

In connection with the test-time adaptation capabilities of our method, methods that combine metric learning with initialization-based models are great competitors. In fact, Proto-MAML (Triantafillou et al., 2019), which captures the best of Prototypical Networks (Snell et al., 2017) and MAML (Finn et al., 2017), allows within-task adaptation using MAML on top of a shared feature extractor. Similarly, Kim et al. (2018) have proposed a Bayesian version of MAML where a feature extractor is shared across tasks, while multiple MAML particles are used for the task-level adaptation. Bertinetto et al. (2018) have also tackled the lack of adaptation for new tasks by using Ridge Regression and Logistic Regression to find the appropriate weighting of the training samples for classification tasks. This study can be considered as an instance of the FSDKL framework, though its contribution was limited to showing that simple differentiable learning algorithms can increase adaptation in the metric learning framework. Our work goes beyond by formalizing few-shot DKL and proposing ADKL: a data-driven manner to compute the correct kernel for a task.

Since ADKL-GP learns task-specific stochastic processes, it is related to neural processes (Garnelo et al., 2018a) and the ML-PIP framework (Gordon et al., 2018). Both propose a scalable alternative to learning regression functions by performing inference on stochastic processes. In these families of methods, both Conditional Neural Processes (CNP) (Garnelo et al., 2018b) and Attentive Neural Processes (ANP) (Kim et al., 2019b) learn conditional stochastic processes parameterized by task-specific conditions derived from the support sets, but CNP is the most related to ADKL-GP. CNP is an instance of ML-PIP when the task encoder gives a point estimate of the task parameters instead of a distribution. Finally, the main differences between ANP and CNP are the architecture of the task-encoder and the lack of mathematical guarantees associated with stochastic processes in CNP (as it does not impose any consistency with respect to a prior process). By comparison, ADKL-GP also learns conditional stochastic processes but has mathematical guarantees thanks to GP and PSD kernels.

## 5 DATASETS

Existing FSR methods have been mostly tested on 1D function regression and pixel-wise image completion tasks with MNIST and CelebA (Kim et al., 2018; Garnelo et al., 2018b;a). On one hand, the 1D regression tasks are all relatively simple, almost noise-less, and homogeneous. On the other hand, methods have been successful for image completion tasks only outside the few-shot regime (i.e. when the number of samples is greater than 500) (Garnelo et al., 2018b;a). For these reasons, we introduce two task collections from a real-world context. Deemed **Binding** and **Antibacterial**, these task collections contain data from bio-assays that are representative of real-world FSR tasks in drug

discovery. The pre-processed versions of these collections and detailed statistics are available here (anonymized link).

**Binding:** All tasks in this collection aim to predict the binding affinity of small molecules to a target protein. The characteristics of the proteins thus define different data distributions over the chemical space. The inputs and the targets for each task are molecules that have been tested in a binding assay and the measured binding affinity of the molecule against a given protein. The task collection was extracted from the public database BindingDB and altered by removing bio-assays with targets correlated above $0.8$ or those with less than 10 experimental measurements, leaving us with $5,717$ tasks.

**Antibacterial:** Within this collection, the task is to predict the antimicrobial activity of small molecules against various bacteria. They are characterized by a bacterial strain whose resistance to drug-like molecules was being evaluated. The task collection was extracted from the public database PubChem. After also removing bio-assays with correlations above $0.8$ and those with less than 10 samples, we obtain $3,255$ tasks.

Their meta-test partitions each contain 1000 tasks, with the remaining used in the meta-train and meta-validation. The molecules (represented as SMILES) are converted into vectors using routines available in the RDKit software (more precisely into ECFP6 binary fingerprint vectors of 4,096 dimensions). These inputs were also processed in all methods using the same feature extractor architecture, which is a fully-connected network of $256 \times 256 \times 256$. Due to the high noise-to-signal ratio, the targets are first $log2$-scaled and then scaled linearly between 0 and 1 to avoid scaling issues during training.

Fig. 2 highlights three aspects of the collections that make them complementary to existing benchmarks, but better suited for evaluating the readiness of FSR methods for real-world applications relative to toy collections. First, the distributions of number of samples per task show that they naturally contain few samples, which we believe reflects the costs of acquiring labelled data in a drug discovery setting. In comparison, the number of samples available per task is relatively large in previous benchmarks, with the few-shot regime being achieved artificially through sampling. Second, as illustrated by their noise-to-signal ratio, real-world tasks are inherently noisy, increasing the difficulties associated with few-shot learning. Finally, the input diversity within each task is reduced relative to the total among tasks. Despite this diversity difference, good models should perform relatively well outside the input region they have seen in the support set. This situation challenges the methods to learn strong priors about the input space and to be able to generalize after seeing only a small fraction of it. These collections invite researchers to explore meta-learning with increasingly heterogeneous datasets and in noisy environments, as well as generalisation and extrapolation in large input spaces (such as the drug-like chemical space, which is estimated to be approximately $10^{33}$ molecules (Polishchuk et al., 2013)).

To test our method in a noise-less environment, we also use the **Sinusoids** collection introduced by Kim et al. (2018). This challenging few-shot regression benchmark consists of 5,000 tasks defined by functions of the form: $y = A \sin(wx + b) + \epsilon$ with $A \in [0.1, 5.0]$, $b \in [0.0, 2\pi]$, and $w \in [0.5, 2.0]$. Sampling inputs $x \in [-5.0, 5.0]$ and observational noise $\epsilon \in N(0, (0.01A)^2)$ and computing $y$ gives the samples for each task. Here, the meta-train, meta-validation, and meta-test contain 56.25%, 18.75% and 25% of all the tasks, respectively, and all methods use the same feature extractor architecture, which is a fully-connected network of $40 \times 40 \times 40$.

# 6 EXPERIMENTS

## 6.1 BENCHMARKING ANALYSIS

For all benchmarks, the performances of ADKL is compared against other meta-learning algorithms: R2-D2 (an instance of FSDKL Bertinetto et al. (2018)), CNP (Garnelo et al., 2018b), MAML (Finn et al., 2017), BMAML (Kim et al., 2018), and Learned Basis (Yi Loo, 2019) (all implementations are available here (anonymized link)). These algorithms have all proven to have efficient and effective test-time adaptation routines and therefore constitute strong baselines for benchmarking. However, for bioassay modelling benchmarks, we have also added two methods considered to be state-of-the-art in chemoinformatics to assess performance relative to all meta-learning approaches. These methods

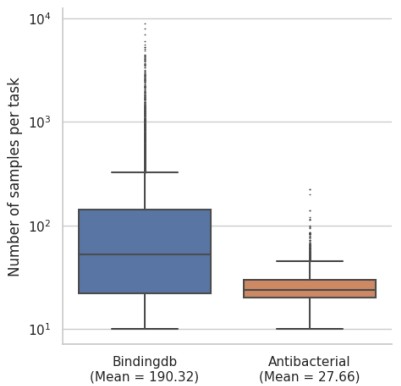 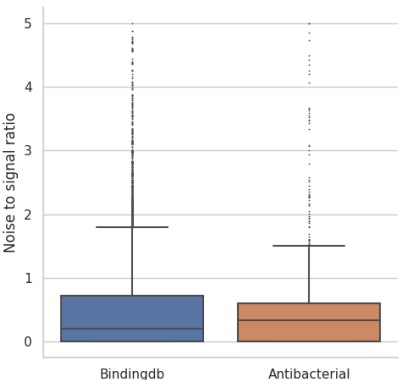 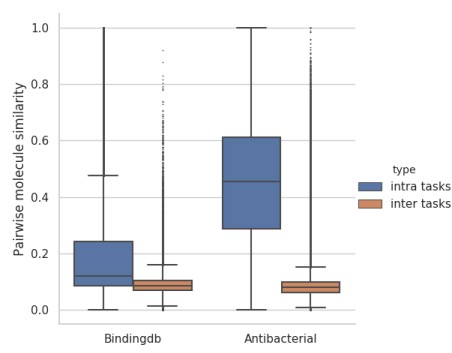

Figure 2: Statistics on bio-assay modelling tasks. Left: Number of samples per task. Middle: Noise-to-signal ratio. Right: Within-task versus overall molecular diversity.

| k model | 5 | 10 | 20 |
|---|---|---|---|
| ADKL-GP | | | |
| ADKL-KRR | $0.0380 \pm 0.0020$ | $0.0348 \pm 0.0009$ | $0.0322 \pm 0.0020$ |
| BMAML | $0.0813 \pm 0.0571$ | $0.0486 \pm 0.0071$ | $0.0487 \pm 0.0012$ |
| CNP | $0.0416 \pm 0.0019$ | $0.0393 \pm 0.0030$ | $0.0397 \pm 0.0027$ |
| ECFP4+KRR | $0.0376 \pm 0.0012$ | $0.0352 \pm 0.0014$ | $0.0317 \pm 0.0016$ |
| ECFP4+RF | $\mathbf{0.0373 \pm 0.0012}$ | $\mathbf{0.0339 \pm 0.0013}$ | $\mathbf{0.0311 \pm 0.0012}$ |
| LearnedBasis | $0.0761 \pm 0.0040$ | $0.0754 \pm 0.0042$ | $0.0616 \pm 0.0215$ |
| R2D2 | $0.0492 \pm 0.0015$ | $0.0460 \pm 0.0110$ | $0.0342 \pm 0.0012$ |

Table 1: Average MSE on Binding

| k model | 5 | 10 | 20 |
|---|---|---|---|
| ADKL-GP | $0.1017 \pm 0.0013$ | $0.0895 \pm 0.0015$ | $\mathbf{0.0860 \pm 0.0016}$ |
| ADKL-KRR | $\mathbf{0.1000 \pm 0.0012}$ | $\mathbf{0.0893 \pm 0.0015}$ | $0.0862 \pm 0.0009$ |
| BMAML | $0.1059 \pm 0.0021$ | $0.1020 \pm 0.0029$ | $0.4616 \pm 0.4210$ |
| CNP | $0.1063 \pm 0.0023$ | $0.1239 \pm 0.0219$ | $0.1382 \pm 0.0049$ |
| ECFP4+KRR | $0.1166 \pm 0.0020$ | $0.1003 \pm 0.0009$ | $0.0956 \pm 0.0009$ |
| ECFP4+RF | $0.1129 \pm 0.0002$ | $0.1016 \pm 0.0008$ | $0.0970 \pm 0.0003$ |
| LearnedBasis | $0.1274 \pm 0.0037$ | $0.1308 \pm 0.0032$ | $0.1329 \pm 0.0043$ |
| R2D2 | $0.1104 \pm 0.0023$ | $0.0962 \pm 0.0021$ | $0.0921 \pm 0.0010$ |

Table 2: Average MSE on Antibacterial

are the Random Forest algorithm with ECFP4 (Extended Connectivity FingerPrints of diameter 4) as molecular input representation, and ECFP4 with KRR and tanimoto similarity as a kernel function (Olier et al., 2018). During meta-test, each task is partitioned into query and support sets, then the support set is used to generate a model which is evaluated on the query set to compute the MSE. This process is repeated 30 times per task and the average MSE over the repetitions per task and over all tasks is reported in Tables 1 to 3.

For the Sinusoids collection, Table 3 shows that DKL-based methods significantly outperform all other methods despite their test-time adaptation capabilities. These results alone demonstrate the effectiveness of DKL-based methods in FSR relative to the current state-of-the-art. Furthermore, of all DKL-based methods, ADKL-KRR shows consistently stronger performance than others. This demonstrates that using ADKL increases test-time performance relative to FS-DKL (as R2-D2 and ADKL-KRR only differ by the kernel definition). It also indicates that attempting to capture the model uncertainty using GP in ADKL (instead of KRR) comes with a significant cost, especially in lower data regimes. This may be due to the inability of GP to differentiate between the observational noise and the model uncertainty as the number of samples get smaller. Also, notice that all task encoding based methods significantly outperform the others. This shows that adequately capturing the task representation is crucial for this task collection, and ADKL-KRR appears to be best equipped to do so.

Tables 1 and 2 show the performances of all methods on real-world datasets. As complements, Tables 4 and 5 show the $p$-value that assesses the statistical significance of the difference between each model and ADKL-GP and ADKL-KRR. These p-values result from Wilcoxon ranked tests comparing the MSE per task of each algorithm to ADKLs. Combined together, these tables shows that ADKL methods significantly outperforms all other meta-learning methods ($p-$values $< 0.05$). They also outperformed the state-of the art in chemoinformatic for Antibacterial, but do not on Binding where those methods are significantly better than all meta-learning algorithms. Even if, ADKL is a first step in the right direction, these results show that there remains much room to develop meta-learning algorithms which are undoubtedly superior to methods in computational chemistry. It is also worth noticing that ADKL methods are significantly better than R2-D2 for these collections also confirming that using task specific kernels are useful and improve generalization.

| m model | 5 | 10 | 20 |
|---|---|---|---|
| BMAML | 2.042 | 1.371 | 0.844 |
| CNP | 1.616 | 0.392 | 0.117 |
| Learned Basis | 3.587 | 0.800 | 0.127 |
| MAML | 2.896 | 1.634 | 0.901 |
| ADKL-GP | 1.178 | 0.084 | 0.007 |
| ADKL-KRR | **0.867** | **0.061** | **0.005** |
| FSDKL (R2D2) | 1.002 | 0.073 | 0.009 |

Table 3: Average MSE on Sinusoidals

| | ADKL-GP | ADKL-KRR |
|---|---|---|
| ADKL-GP | | 1.21e-01 |
| CNP | 0.00e+00 | 0.00e+00 |
| ADKL-KRR | 1.21e-01 | |
| ECFP4+KRR | 2.48e-78 | 3.69e-62 |
| LearnedBasis | 0.00e+00 | 0.00e+00 |
| BMAML | 0.00e+00 | 0.00e+00 |
| FSDKL (R2D2) | 1.18e-41 | 5.50e-15 |
| ECFP4+RF | 9.70e-81 | 2.20e-168 |

Table 4: Wilcoxon p-values – BindingDB

| | ADKL-KRR |
|---|---|
| ADKL-GP | |
| CNP | 1.77e-114 |
| ADKL-KRR | |
| ECFP4+KRR | 2.70e-03 |
| LearnedBasis | 0.00e+00 |
| BMAML | 0.00e+00 |
| FSDKL (R2D2) | 1.76e-49 |
| ECFP4+RF | 5.94e-01 |

Table 5: Wilcoxon p-values – BindingDB

## 6.2 ACTIVE LEARNING

In this section, we report the results of active learning experiments. Our intent is to measure the effectiveness of the uncertainty captured by the predictive distribution of ADKL-GP for active learning, as it is critical to our drug discovery use-cases. CNP, in comparison, serves to measure which of CNP and GP better captures the data uncertainty for improving FSR under active sample selection. For this purpose, we meta-train both algorithms using support and query sets of size $m = 5$. During meta-test time, five samples are randomly selected to constitute the support set $D_{trn}$ and build the initial hypothesis for each task. Then, from a pool $U$ of unlabeled data, we choose the input $\mathbf{x}^*$ of maximum predictive entropy, i.e. $\mathbf{x}^* = \mathrm{argmax}_{\mathbf{x} \in U} \mathbb{E}\left[\log p(y|\mathbf{x}, D_{trn})\right]$. The latter is removed from $U$ and added to $D_{trn}$ with its predicted label. The within-task adaptation is performed on the new support set to obtain a new hypothesis which is evaluated on the query set $D_{val}$ of the task. This process is repeated until we reach the allotted budget of 20 queries.

Fig. 3 illustrates, for all collections, the MSE after each sample acquisition iteration and under both random and active learning acquisition strategies. Under the active learning strategy, ADKL-GP consistently outperforms CNP. In particular, we observe that very few samples are queried by ADKL-GP to capture the data distribution whereas CNP performance remains far from optimal even when allowed the maximum number of queries. Further, since using the maximum predictive entropy strategy is better than querying samples at random for ADKL-GP (solid vs. dashed line), these results suggest that the predictive uncertainty obtained with GP is informative and more accurate than that of CNP. Moreover, when the number of queries is greater than 10, we observe a performance degradation for CNP while ADKL-GP remains consistent. This observation highlights the generalization capacity of DKL methods, even outside the few-shot regime where they have been trained — this same property does not hold true for CNP. We attribute this property of DKL methods to their use of kernel methods. In fact, their role in adaptation and generalization increases as we move away from the few-shot training regime.

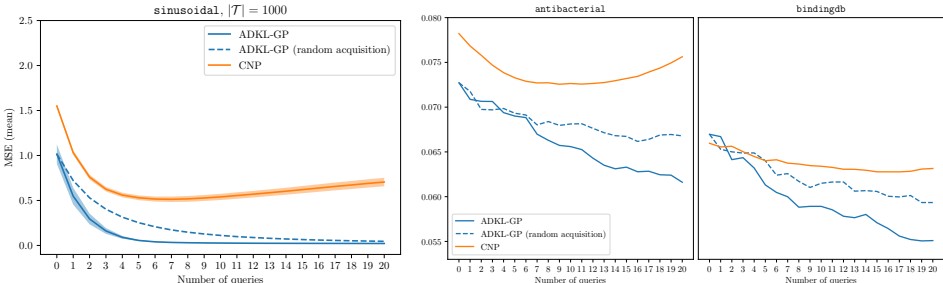

Figure 3: Average MSE performance on the meta-test during active learning. The width of the shaded regions denotes the uncertainty over five runs for the sinusoidal collection. No uncertainty is shown for the real-world tasks as they were too time consuming.

## 6.3 ABLATION EXPERIMENTS

In our final set of experiments, we more closely evaluate the impact of the task encoder and the pseudo-inputs on the generalization during meta-testing. We do so by training and evaluating ADKL on

347 Sinusoids with different hyperparameter combinations. Figs. 4a to 4d show the relative improvements
348 (negative values) or setbacks (positive values) in the meta-test MSE compared to different baselines
349 (but the joint impact of $\gamma_{task}$ and $\gamma_{pseudo}$ is only discussed in Appendix A.3).

350 First, Fig. 4a compares $\gamma_{task} \in \{0.01, 0.1\}$ relative to $\gamma_{task} = 0$ and consequently demonstrates that
351 regularizing the task encoder by maximizing the mutual information between the support set and
352 the query set significantly improves the generalization performance. This conclusion holds for all
353 support set sizes tested, as shown in Appendix A.1. Combined with the results from Section 6.1, this
354 figure shows the importance of good task encoders for generalization in few-shot learning and how
355 using the regularization term that we introduced is a step forward in that direction.

356 Then, Fig. 4c measures the relative differences between $\gamma_{pseudo} \in \{0.01, 0.1\}$ and $\gamma_{pseudo} = 0$ for
357 different values of hyperparameter combinations. It shows that improving the kernel map evaluations
358 using *pseudo-input representations* can significantly help with the generalization performance of
359 ADKL. This conclusion also holds for all values tested for $|D_{trn}^t|$ ( see Appendix A.2). However, the
360 improvements were more consistent for smaller support sets, which is not surprising as improving
361 the kernel map estimations in these cases is more critical.

362 Finally, Figs. 4b and 4d illustrate for ADKL-GP and ADKL-KRR, and different sizes of support sets,
363 how the number of pseudo-representations (i.e $|U|$) affects performance. The values for each cell
364 are relative performance using $|U| \in \{20, 50\}$ versus $|U| = 0$ and have been averaged over different
365 hyperparameters and $\gamma_{pseudo}$. In general, we can confirm that increasing the number of pseudo-
366 representations increases the estimates of the kernel maps and improves generalization. However, the
367 improvements are more prominent with KRR in comparison to GP, which may be due to the fact that
368 GP attributes a part of the modelling noise to the kernel evaluations, leading to more constraints on
369 the optimization of the pseudo-representation parameters.

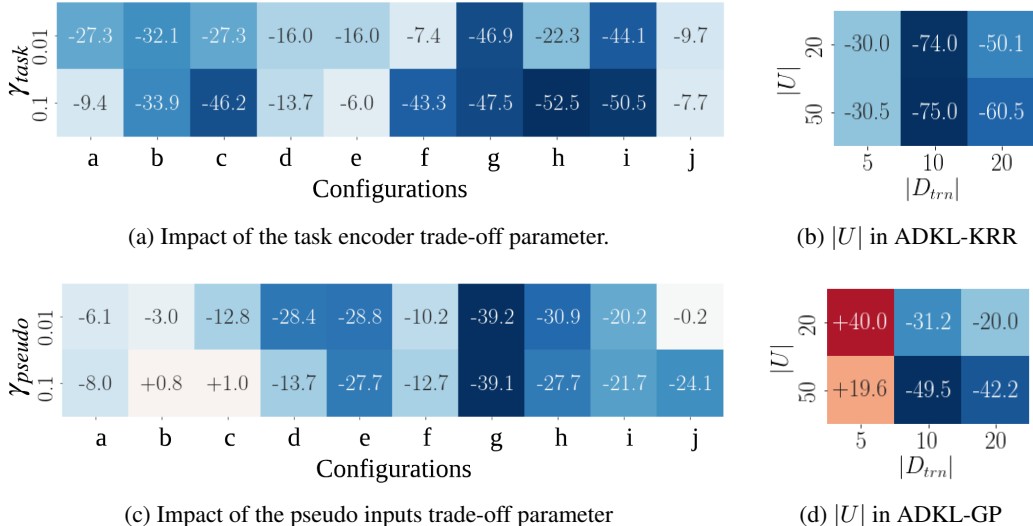

(a) Impact of the task encoder trade-off parameter.

(b) $|U|$ in ADKL-KRR

(c) Impact of the pseudo inputs trade-off parameter

(d) $|U|$ in ADKL-GP

Figure 4: Relative decrease/increase in the meta-test MSE compared to different baselines. In (a) and
(c) the baselines are $\gamma_{task} = 0$ and $\gamma_{pseudo} = 0$, respectively. In (b) and (d) the baselines are $|U| = 0$

## 7 CONCLUSION

371 We investigate bio-assays modelling using FSR methods. Our proposed method, ADKL, stores
372 meta-knowledge in kernel functions and adapts to new tasks using KRR or GP. Our experiments
373 provide evidence that the additional adaptation capacity at test-time provided by ADKL increases
374 generalization significantly. Also, in a Bayesian setup, ADKL provides better predictive uncertainty,
375 increasing their utility in bioassay modelling. However, there is still room to improve ADKL and
376 most meta-learning methods to be better than traditional chemoinformatic methods. We hope, by
377 making our bio-assay task collections publicly available, that the community will leverage them to
378 propose new competitive FSR methods.

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

# Appendices

468

## A REGULARIZATION IMPACT

469

### A.1 TASK REGULARIZATION

470

471 Table 6 presents the hyperparameter combinations used in the experiments to assess the impact of
472 the trade-off parameter $\gamma_{task}$. We report the MSE performance obtained on the meta-test for each
473 combination. To make reading this table easier, we also repeat the Fig. 5 showing the improvement
474 of the MSE relative to $\gamma_{task} = 0$ (no regularization).

Table 6: Effect of using task regularization (parameter $\gamma_{task}$) on the MSE performance

| algorithm | K | $\gamma_{pseudo}$ | $\gamma_{task}$ Configuration | 0.00 | 0.01 | 0.10 |
|---|---|---|---|---|---|---|
| ADKL-KRR | 20 | 0.01 | a | 0.0585 | 0.0327 | **0.0289** |
| | 10 | 0.00 | b | 0.4051 | **0.2944** | 0.3671 |
| | | 0.10 | c | 0.4363 | 0.2964 | **0.2882** |
| ADKL-GP | 5 | 0.10 | d | 2.4920 | **2.2511** | 2.2994 |
| ADKL-KRR | 20 | 0.00 | e | 0.0574 | 0.0305 | **0.0302** |
| ADKL-GP | 5 | 0.01 | f | 2.5611 | **2.1511** | 2.2112 |
| | | 0.01 | g | 3.2933 | **2.7663** | 3.0971 |
| | 10 | 0.01 | h | 0.7675 | 0.7105 | **0.4352** |
| | 20 | 0.00 | i | 0.1201 | 0.0873 | **0.0646** |
| ADKL-KRR | 20 | 0.10 | j | 0.0575 | 0.0447 | **0.0273** |

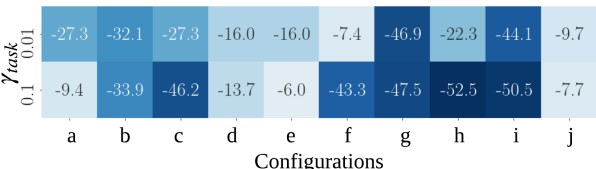

Figure 5: Relative improvement of the MSE depending on the $\gamma_{task}$ parameter

475 For a more in-depth analysis, we show below the similar tables and figures for different values of $K$ (
476 5, 10 and 20). These results confirm that regularizing the task encoder is helpful for any value of
477 $K$, even though the impact seems to become much more important as $K$ increases (observe that the
478 maximum improvement in each figure increases with $K$).

479 **For $K = 5$**

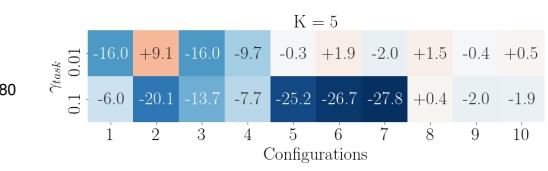

480

| algorithm | $\gamma_{pseudo}$ | $\gamma_{task}$ 0.00 | 0.01 | 0.10 |
|---|---|---|---|---|
| ADKL-GP | 0.01 | 3.2933 | 2.7663 | 3.0971 |
| | 0.00 | 2.8528 | 3.1136 | 2.2801 |
| | 0.01 | 2.5611 | 2.1511 | 2.2112 |
| | 0.10 | 2.4920 | 2.2511 | 2.2994 |
| ADKL-KRR | 0.00 | 1.7123 | 1.7079 | 1.2808 |
| | 0.01 | 1.6344 | 1.6655 | 1.1974 |
| | 0.10 | 1.6868 | 1.6532 | 1.2173 |
| | 0.00 | 1.1951 | 1.2129 | 1.1998 |
| | 0.01 | 1.1655 | 1.1611 | 1.1416 |
| | 0.10 | 1.1658 | 1.1716 | 1.1442 |

481 **For $K = 10$**

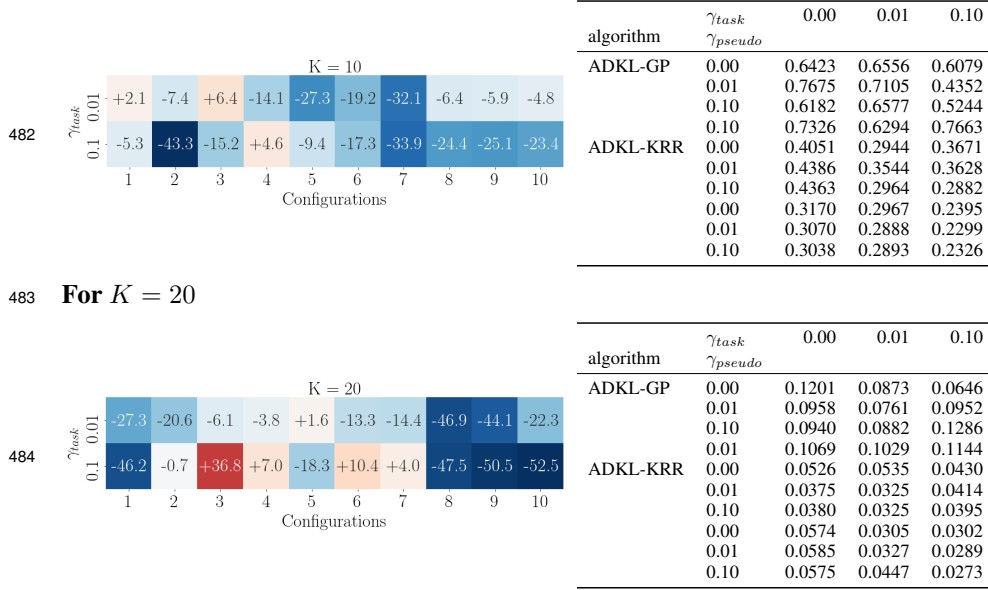

| algorithm | $\gamma_{pseudo}$ | $\gamma_{task}$ 0.00 | 0.01 | 0.10 |
|---|---|---|---|---|
| ADKL-GP | 0.00 | 0.6423 | 0.6556 | 0.6079 |
| | 0.01 | 0.7675 | 0.7105 | 0.4352 |
| | 0.10 | 0.6182 | 0.6577 | 0.5244 |
| | 0.10 | 0.7326 | 0.6294 | 0.7663 |
| ADKL-KRR | 0.00 | 0.4051 | 0.2944 | 0.3671 |
| | 0.01 | 0.4386 | 0.3544 | 0.3628 |
| | 0.10 | 0.4363 | 0.2964 | 0.2882 |
| | 0.00 | 0.3170 | 0.2967 | 0.2395 |
| | 0.01 | 0.3070 | 0.2888 | 0.2299 |
| | 0.10 | 0.3038 | 0.2893 | 0.2326 |

**For $K = 20$**

| algorithm | $\gamma_{pseudo}$ | $\gamma_{task}$ 0.00 | 0.01 | 0.10 |
|---|---|---|---|---|
| ADKL-GP | 0.00 | 0.1201 | 0.0873 | 0.0646 |
| | 0.01 | 0.0958 | 0.0761 | 0.0952 |
| | 0.10 | 0.0940 | 0.0882 | 0.1286 |
| | 0.01 | 0.1069 | 0.1029 | 0.1144 |
| ADKL-KRR | 0.00 | 0.0526 | 0.0535 | 0.0430 |
| | 0.01 | 0.0375 | 0.0325 | 0.0414 |
| | 0.10 | 0.0380 | 0.0325 | 0.0395 |
| | 0.00 | 0.0574 | 0.0305 | 0.0302 |
| | 0.01 | 0.0585 | 0.0327 | 0.0289 |
| | 0.10 | 0.0575 | 0.0447 | 0.0273 |

## A.2 PSEUDO-INPUT REPRESENTATIONS

Table 7 presents the hyperparameter combinations used in the experiments to assess the impact of the trade-off parameter $\gamma_{pseudo}$, which governs the penalty applied to the divergence between the distribution of learned pseudo-representations and the distribution of actual representations. We also repeat in Fig. 6, the relative improvement of MSE compared to $\gamma_{pseudo} = 0$ as shown in the main text.

Table 7: Effect of the pseudo-examples regularization (parameter $\gamma_{pseudo}$) on the MSE performance

| algorithm | K | $\gamma_{task}$ | $\gamma_{pseudo}$ Conf. | 0.00 | 0.01 | 0.10 |
|---|---|---|---|---|---|---|
| ADKL-GP | 10 | 0.10 | a | 0.6079 | **0.4352** | 0.5244 |
| | 20 | 0.01 | b | 0.0873 | **0.0761** | 0.0882 |
| ADKL-KRR | 20 | 0.00 | c | 0.0526 | **0.0375** | 0.0380 |
| ADKL-GP | 5 | 0.10 | d | 2.2801 | **2.2112** | 2.2994 |
| ADKL-KRR | 20 | 0.01 | e | 0.0535 | **0.0325** | **0.0325** |
| ADKL-GP | 5 | 0.01 | f | 2.9466 | 2.7663 | **2.7121** |
| | 20 | 0.10 | g | 0.1147 | 0.1144 | **0.0870** |
| | | 0.00 | h | 0.1201 | 0.0958 | **0.0940** |
| | 5 | 0.01 | i | 3.1136 | **2.1511** | 2.2511 |
| | | 0.00 | j | 2.8528 | 2.5611 | **2.4920** |

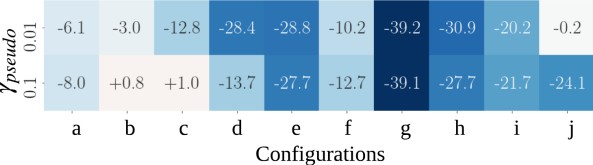

Figure 6: Relative improvement of the MSE depending on the $\gamma_{task}$ parameter

Once again, for a more in-depth analysis, we show below the same format of tables and figures for different values of $K$, confirming again that regularizing using the pseudo-representation can be very helpful for any value of $K$. It is worth noticing here that the improvement gain is more consistent for

$K = 5$ compared to $K \in \{10, 20\}$, supporting the fact that improving kernel maps evaluations using pseudo-representations is critical as size of the support set decreases.

**For $K = 5$**

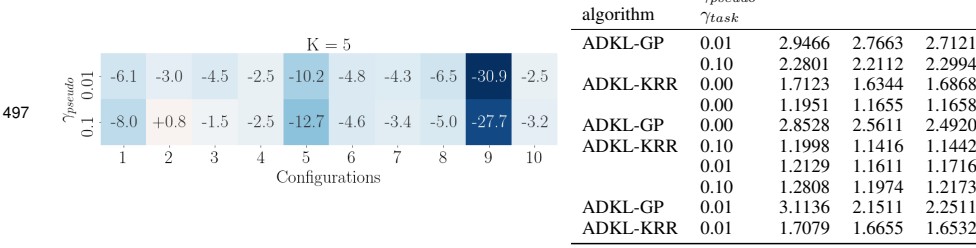

| algorithm | $\gamma_{pseudo}$ $\gamma_{task}$ | 0.00 | 0.01 | 0.10 |
|---|---|---|---|---|
| ADKL-GP | 0.01 | 2.9466 | 2.7663 | 2.7121 |
|  | 0.10 | 2.2801 | 2.2112 | 2.2994 |
| ADKL-KRR | 0.00 | 1.7123 | 1.6344 | 1.6868 |
|  | 0.00 | 1.1951 | 1.1655 | 1.1658 |
| ADKL-GP | 0.00 | 2.8528 | 2.5611 | 2.4920 |
| ADKL-KRR | 0.10 | 1.1998 | 1.1416 | 1.1442 |
|  | 0.01 | 1.2129 | 1.1611 | 1.1716 |
|  | 0.10 | 1.2808 | 1.1974 | 1.2173 |
| ADKL-GP | 0.01 | 3.1136 | 2.1511 | 2.2511 |
| ADKL-KRR | 0.01 | 1.7079 | 1.6655 | 1.6532 |

**For $K = 10$**

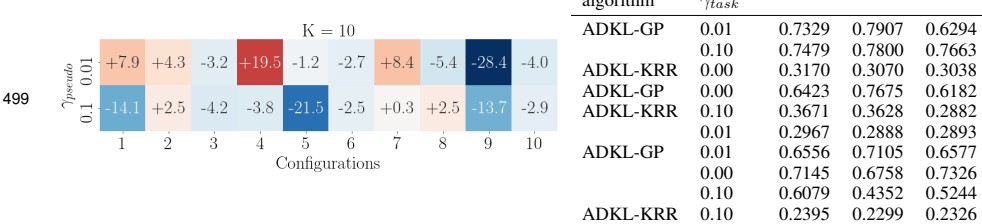

| algorithm | $\gamma_{pseudo}$ $\gamma_{task}$ | 0.00 | 0.01 | 0.10 |
|---|---|---|---|---|
| ADKL-GP | 0.01 | 0.7329 | 0.7907 | 0.6294 |
|  | 0.10 | 0.7479 | 0.7800 | 0.7663 |
| ADKL-KRR | 0.00 | 0.3170 | 0.3070 | 0.3038 |
| ADKL-GP | 0.00 | 0.6423 | 0.7675 | 0.6182 |
| ADKL-KRR | 0.10 | 0.3671 | 0.3628 | 0.2882 |
|  | 0.01 | 0.2967 | 0.2888 | 0.2893 |
| ADKL-GP | 0.01 | 0.6556 | 0.7105 | 0.6577 |
|  | 0.00 | 0.7145 | 0.6758 | 0.7326 |
|  | 0.10 | 0.6079 | 0.4352 | 0.5244 |
| ADKL-KRR | 0.10 | 0.2395 | 0.2299 | 0.2326 |

**For $K = 20$**

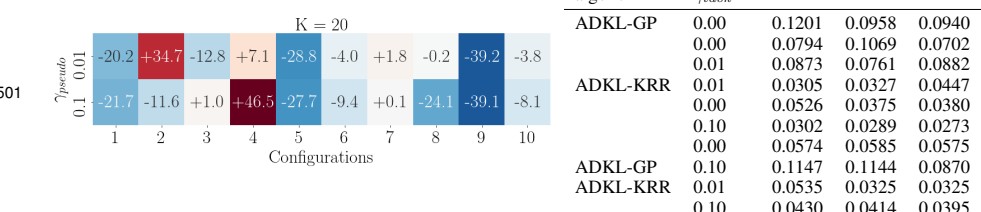

| algorithm | $\gamma_{pseudo}$ $\gamma_{task}$ | 0.00 | 0.01 | 0.10 |
|---|---|---|---|---|
| ADKL-GP | 0.00 | 0.1201 | 0.0958 | 0.0940 |
|  | 0.00 | 0.0794 | 0.1069 | 0.0702 |
|  | 0.01 | 0.0873 | 0.0761 | 0.0882 |
| ADKL-KRR | 0.01 | 0.0305 | 0.0327 | 0.0447 |
|  | 0.00 | 0.0526 | 0.0375 | 0.0380 |
|  | 0.10 | 0.0302 | 0.0289 | 0.0273 |
|  | 0.00 | 0.0574 | 0.0585 | 0.0575 |
| ADKL-GP | 0.10 | 0.1147 | 0.1144 | 0.0870 |
| ADKL-KRR | 0.01 | 0.0535 | 0.0325 | 0.0325 |
|  | 0.10 | 0.0430 | 0.0414 | 0.0395 |

Overall, the effect of the regularization is beneficial, even though we witness a few pathological cases.

### A.3   JOINT IMPACT OF $\gamma_{task}$ AND $\gamma_{pseudo}$

Since both $\gamma_{task}$ and $\gamma_{pseudo}$ have a high impact on the training and the generalization performance, we need to assess the relationship between the two. Fig. 7 shows, for different values of $K$, the relative improvement of the test MSE compared to the case where no regularization is done, i.e. $\gamma_{task} = 0$ and $\gamma_{pseudo} = 0$. Overall, one can see that higher is better in both dimensions but there seems to be a sweet spot on the grid for each value of $K$ and therefore we can only advise the user to cross-validate on those hyperparameters.

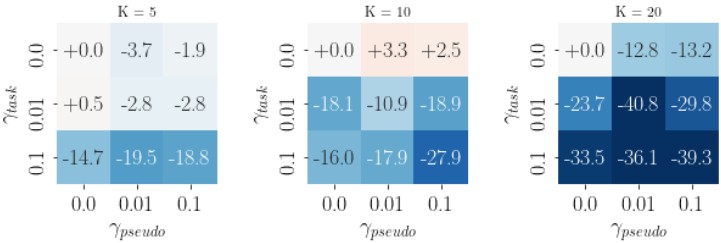

Figure 7: Average relative improvement of the MSE
and joint impact of $\gamma_{task}$ and $\gamma_{pseudo}$.

## B PREDICTION CURVES ON THE SINUSOIDS COLLECTION

Figure 8 presents a visualization of the results obtained by each model on three tasks taken randomly from the meta-test set. We provide the model with ten examples from an unseen task consisting of a slightly noisy sine function (shown in blue), and present in orange the predictions made by the network based on these ten examples.

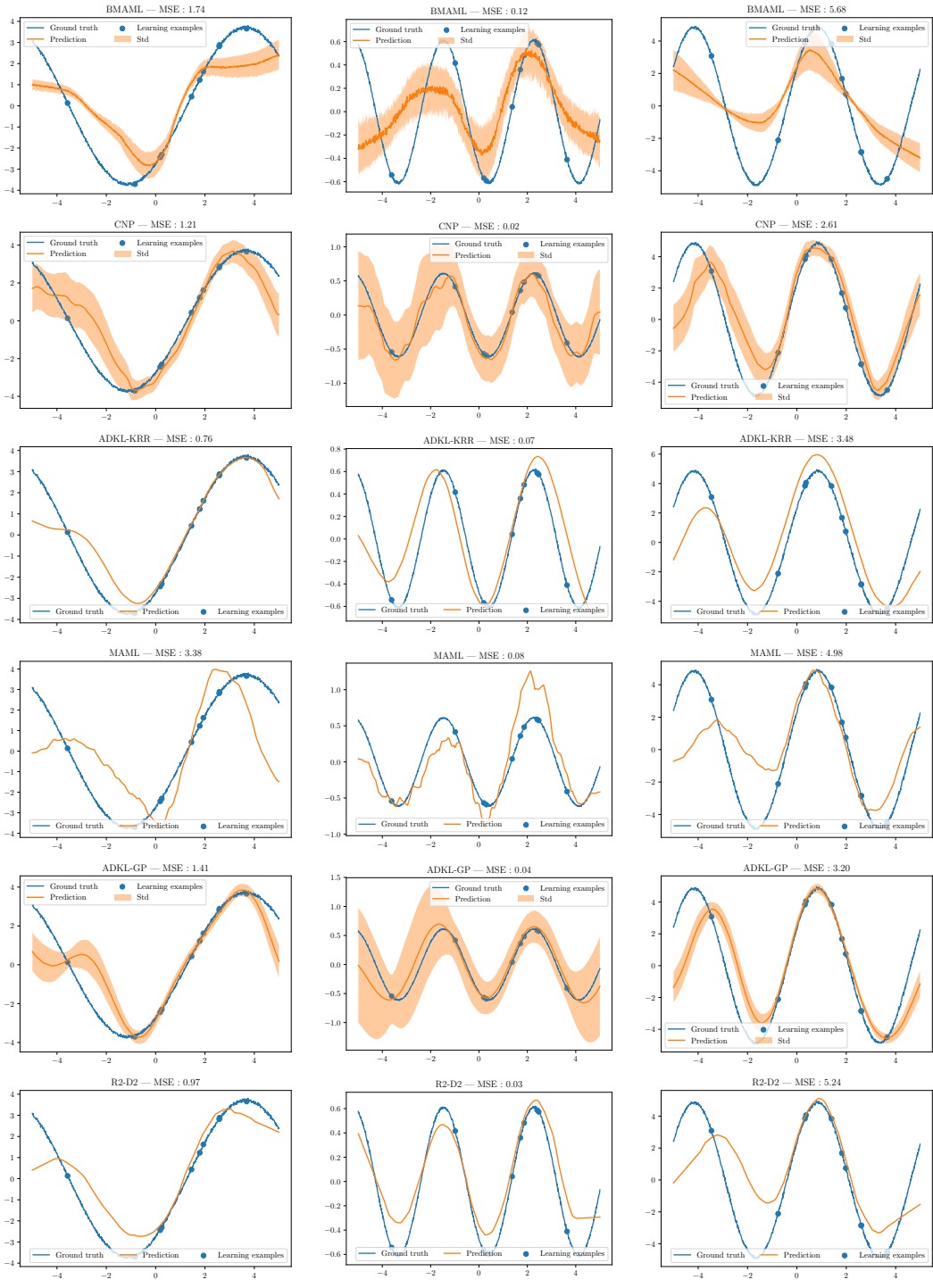

Figure 8: Meta-test time predictions on the `Sinusoids` collection

## C    SUPPLEMENTARY RESULTS FOR THE REAL-WORLD DATASETS

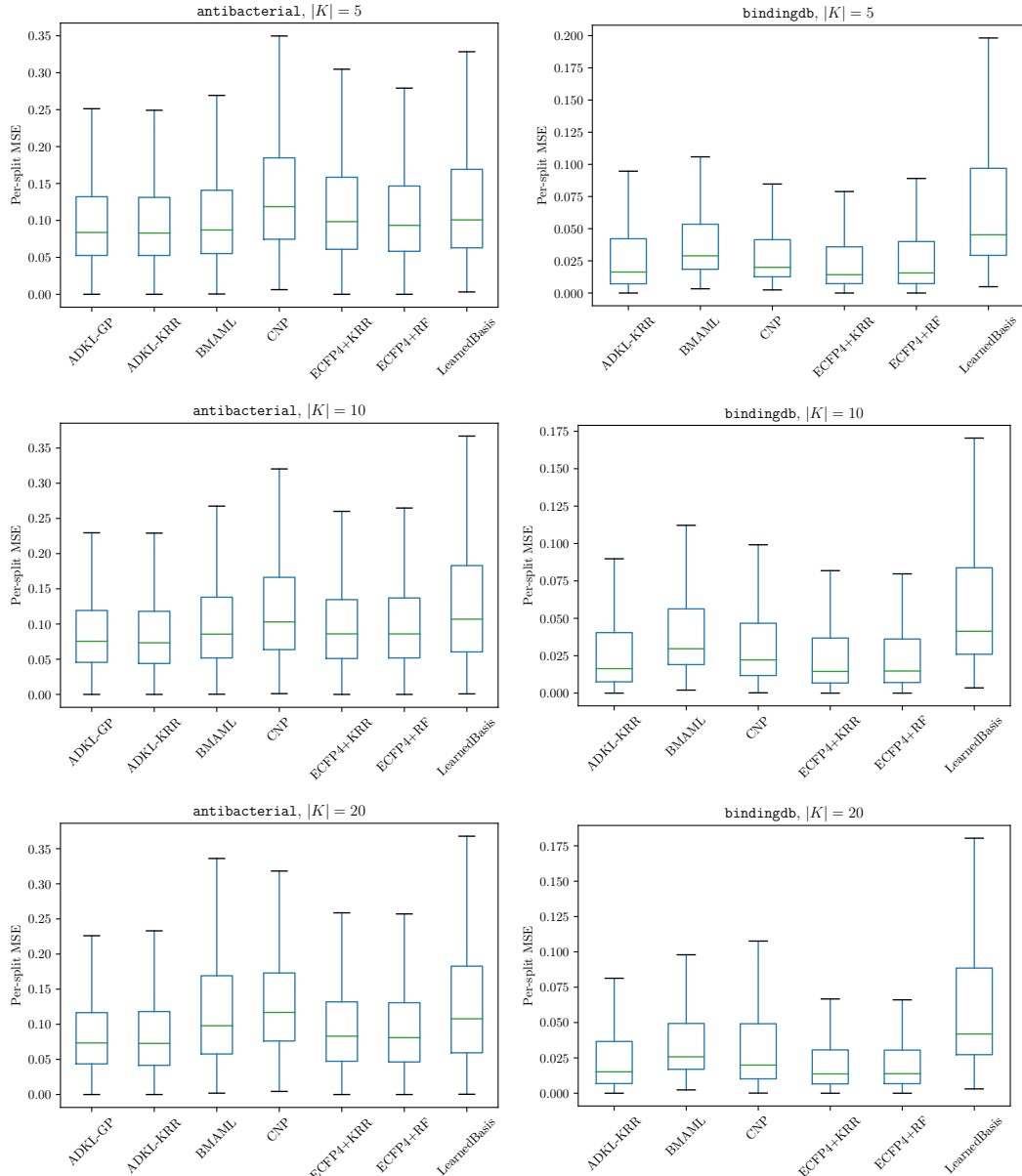

Figure 9: Distribution of the mean squared error (MSE) across the tasks

Figure 9 shows the distribution over the random support/query sets generated at test time. Note that the results presented in the main paper estimate the influence of the initialisation by using multiple seeds and computing the standard deviation on the *average MSE* (averaged over the support/query splits).

The two pieces of information are important : the results presented above give us a better idea of the "*meta-generalisation*" capabilities of each algorithm, while those in the main paper assess the reproducibility and the statistical significance of the relative improvements.

