# OpenReview forum: "MODELLING   BIOLOGICAL   ASSAYS   WITH ADAPTIVE DEEP KERNEL LEARNING"
_ICLR.cc/2020/Conference — Reject_

### Official Review · AnonReviewer7 · 2019-10-31
**Official Blind Review #7**

**Rating:** 6

**Review:**

The authors of this work are attempting to solve the problem of few shot regression for drug discovery problems. This is an important problem in the field of deep drug discovery, since low data issues are very common. While previous papers have addressed the challenges of low data drug classification, there hasn't been much progress made so far on low data regression thus far.

The authors propose formulating the problem at hand as a few shot deep kernel regression problem (FSDKL). This framework has some similarities to past work deep metric learning, which has been used previously for few shot classification problems in drug discovery. In particular, the authors propose a new method, adaptive deep kernel learning to help improve task specific learning. ADKL is claimed to improve over FSDKL since it helps learn a task specific kernel function. There are a number of interesting methodological adaptations here, such as the use of an idea similar to DeepSets to encode order invariance in the training set (an important challenge when dealing with small "support" sets in training). The authors also incorporate some unlabeled data during training to improve on representation learning.

To test their contributions, the authors gather two new dataset collections, Binding and Antibacterial from publicly available sources. Some detail is provided about these collections, but a priori, it's not easy for me to judge the quality of these data sources since they haven't been benchmarked previously in the literature.

In the experimental section, the authors compare ADKL against a number of past low data learning methods. The benchmarks in section 6.1 would benefit from confidence intervals. In my experience, few-shot algorithms can be quite noisy, so statistical tests are important to distinguish architectural improvements from noise. Repeated trials could be taken over the choice of random seed for the experiments to gauge robustness. Without error bars, and on a new dataset, it's not possible to gauge if ADKL really provides an improvement.

The experiments having to do with active learning in section 6.2 are interesting. Why is only CNP measured though? Do other methods not make sense with active learning?

In conclusion, the authors consider an important problem for deep learning in drug discovery and offer a useful advance by adapting the framework of deep kernel learning to this space. However, the work could still use a good bit of polish to really shine. It's not clear to me that the suggested ADKL framework really makes an improvement over past few shot regression methods. Adding some statistical significance tests would be useful. It would also help if the authors benchmarked against datasets that were better known in this field, such as those from the moleculenet.ai suite. It's not easy to judge how easy/hard the new datasets the authors propose are, which makes it challenging to gauge the real improvement. I'm also not sure that this paper is a clear fit for ICLR. I think there's a real contribution to the field of deep drug discovery by the adaptation of the DKL framework to few shot drug discovery, but I'm not convinced that ADKL is superior to other DKL methods as a pure learning technique.

All that said, I'm comfortable marking the paper as a "weak accept" since I think there is a real scientific contribution here, but I encourage the authors to make a serious effort to improve their presentation and tighten-up their experiments.

Detailed Suggestions:
- Section 2 on Deep Kernel Learning and Section 4 on Related work should likely be merged together. Going back and forth between the literature and author contributions was a little confusing.
- There are a lot of acronyms in the paper; I found myself having to refer back and forth to all the methods considered. It might be worthwhile to make a cleanup pass to make the discussions a little easier to read.
- Figure 4 would benefit from a legend. At first glance, it wasn't clear what gamma_mine and gamma_pseudo mean here. The color plot is also a little difficult to make sense. Perhaps consider an alternative plot.

**Experience Assessment:**

I have published in this field for several years.

**Review Assessment: Checking Correctness Of Derivations And Theory:**

I assessed the sensibility of the derivations and theory.

**Review Assessment: Checking Correctness Of Experiments:**

I assessed the sensibility of the experiments.

**Review Assessment: Thoroughness In Paper Reading:**

I read the paper at least twice and used my best judgement in assessing the paper.

---

> ### Author Response · Authors · 2019-11-15
> **response to #7**
>
> Thank you for your thorough review. All comments and questions were very much appreciated and we hope this response addresses them in a way that is satisfactory.
>
> Q1 - The benchmarks in section 6.1 would benefit from confidence intervals. In my experience, few-shot algorithms can be quite noisy, so statistical tests are important to distinguish architectural improvements from noise. Repeated trials could be taken over the choice of random seed for the experiments to gauge robustness. Without error bars, and on a new dataset, it's not possible to gauge if ADKL really provides an improvement.
> R1 - You are absolutely right - without repeated trials the result bears little significance. We have repeated these experiments 3 times each (and apologize for not being able to do more due to time and computational resource constraints) and have updated the manuscript with the new results.
>
> Q2 - The experiments having to do with active learning in section 6.2 are interesting. Why is only CNP measured though? Do other methods not make sense with active learning?
> R2 - Only CNP and BMAML make sense for the active learning setting. However, we did not include  BMAML  because it only gives good results for large number of particles  and requires some monte carlo sampling to compute the variance. So we only used CNP due to computational resource limitations.
>
> Q3 - However, the work could still use a good bit of polish to really shine.
> R3 - You are absolutely right that some polishing was required to improve the paper. As mentioned in the comment addressed to all reviewers, we have performed new experiments and made changes to the text to improve both the technical content and the overall presentation of the paper.
>
> Q4 - It's not clear to me that the suggested ADKL framework really makes an improvement over past few shot regression methods. Adding some statistical significance tests would be useful.
> R4 - We have updated the experiment set performed on real-world dataset to make the meta-test include 1000 tasks, and added some statistical significance tests between all the models and ADKL-KRR. With these results we can see that ADKL-KRR is statistically superior to existing meta-learning methods and is competitive with the state of the art in chemoinformatic
>
> Q5 - It would also help if the authors benchmarked against datasets that were better known in this field, such as those from the moleculenet.ai suite. It's not easy to judge how easy/hard the new datasets the authors propose are, which makes it challenging to gauge the real improvement.
> R5 - Using Moleculenet.ai is certainly a great idea, but only 50 regression tasks are present in the suite and all of them contain large amounts of data (making them less representative of many real-life tasks encountered in bioassay modelling). Due to time constraints, we did not perform these experiments, but it is worth noting that since no previous work has used this suite in a few-shot learning setting, it would still be difficult to gauge the improvement of methods and the difficulty of the tasks. That said, we complemented Tables 2 and 3 with standard deviation of the MSE over multiple runs. We also added statistical tests showing how other  compared to ADKL. We also include in the benchmark the state of the art methods used in chemoinformatics to help gauge the difficulty of the task collection for meta-learning methods.
>
> Q6 - I'm also not sure that this paper is a clear fit for ICLR. I think there's a real contribution to the field of deep drug discovery by the adaptation of the DKL framework to few shot drug discovery,
> Due to the interdisciplinary nature of this work, it is hard to completely disagree with you. However, given that we are proposing a new few-shot learning algorithm, it may also be difficult for the drug discovery and chemoinformatics community to fully appreciate the technical contribution herein. We believe that the ICRL/machine learning community is better suited to appreciate the technical content of this paper, and we are planning to supplement the manuscript with additional materials such as blog posts to help better communicate these technical contribution to the drug discovery community.
>
> Q7 - but I'm not convinced that ADKL is superior to other DKL methods as a pure learning technique.
> R7 - In the updated manuscript, we have provided confidence intervals over the results and statistical test results to help assess whether ADKL is superior to other methods (including other DKL methods). Together, we hope that this information will help illustrate that ADKL is significantly different from compared to other meta-learning baseline including FS-DKL.

---

### Official Review · AnonReviewer5 · 2019-11-01
**Official Blind Review #5**

**Rating:** 3

**Review:**

To tackle data scarcity issue in biological assay modeling, this paper proposes a method to episodically train Deep Kernel Learning models such that at meta-test time the models requires less examples.

The proposed Adaptive Deep Kernel Learning extends Deep Kernel Learning (end-to-end train Gaussian Process/Kernel Ridge Regression using Neural Network's final layer embedding) by:

1. using a learnable task-specific kernel generator network
2. applying episodic few-shot/meta learning to train the system end-to-end

Here are the major issues for rejecting:

Model:
- Section 3.1 and 3.2 about "Task Specific Kernel" are very hard to follow.
  -- (line 170) equation 12, What is the size of the MLP that is used as the kernel network
  -- Is there an ablation to show the contribution of using learnable kernel network vs hand-picked kernel for ADKL-GP and ADKL-KRR?
  -- (line 181) "where U is a set of unlabeled inputs" Where does the unlabeled inputs come from?
  -- Is there an ablation about |U|  <= 50 and its effect?

Experiments and Datasets:

- The ProtoMAML code provided by the authors has only a few lines of code. And in the comment, it says: "It turns out that ProtoMAML is the same as the light version of BMAML with one particle." Such implementation detail should be mentioned in the main paper.

- Section 5 lacks details.
 -- How each meta-train episode is formed? Did they stochasticall sample a subset of molecules for each episode?
 -- What is the total number of tasks in the meta-train and meta-test split respectively? Are there any overlapping between the two splits?


Presentation Issues:
- Section 2 (line 76) "We extended it (Deep Kernel Learning) to few-shot learning and discuss its advantages over the metric learning framework" This sounds like the author developed a brand new framework, while the rest of paper is about proposing a specific realization of few-shot learning: Few-shot Deep Kernel Learning, which meta-learn through a differentialble kernel learning process (with the task kernel adaptation network novelty).

- Section 3 (Figure 1) "The blue and orange colors show the procedure..." However there are more than two colors (including different shades of blue and orange colors) in the figure, which makes the figure hard to parse .

- Section 4 (line 201) "DKL methods lie between neural networks and kernel methods". To me, at high-level, DKL adds kernel learning as a differentiable layer in the end of a neural network. See how a similar model is categorized in Bertinetto et al. ICLR 2019 paper "Meta-learning with differentiable closed-form solvers":  "The main idea is to teach a deep network to use standard machine learning tools, such as ridge regression,as part of its own internal model, enabling it to quickly adapt to novel data." The delta of the paper is adding 1) Gaussian Process and 2) task-specific adaptation. If Bertinetto et al. categorize their method as part of a deep network then I think so should this work.

- Section 4 (line 239) "Our work goes beyond ... by proposing ADKL: a data-driven manner for computing the correct kernel for a task." This is probably a grammar mistake. "a data driven manner for" sounds odd.

- Section 5 (line 272) "Fig 2 highlights three aspects of the collections that make them better benchmarks for evaluating the readiness of FSR methods for real-world applications relative to toy collections." What is the other benchmarks the authors are comparing with? Are they generic benchmarks for all types few-shot regression tasks/applications beyond biological assays (e.g. computer vision tasks like: object detection). Maybe defining the proposed benchmark as "complimentary" and "bio-assay application focused" would be more appropriate.



**Experience Assessment:**

I have published one or two papers in this area.

**Review Assessment: Checking Correctness Of Derivations And Theory:**

I assessed the sensibility of the derivations and theory.

**Review Assessment: Checking Correctness Of Experiments:**

I assessed the sensibility of the experiments.

**Review Assessment: Thoroughness In Paper Reading:**

I read the paper at least twice and used my best judgement in assessing the paper.

---

> ### Author Response · Authors · 2019-11-15
> **response to #5 -- part 1**
>
> We sincerely thank you for your thorough review. It really helped us improve many aspects of the paper. We hope to answer all your comments in the remainder of this response.
>
> Q1 - Section 3.1 and 3.2 about "Task Specific Kernel" are very hard to follow.
> R1 - We have re-written these sections and hope that any misleading and confusing parts are now clarified.
>
> Q2 - (line 170) equation 12, What is the size of the MLP that is used as the kernel network
> R2 - We used a two layers MLP of 256 * 1 with ReLU activation in our experiments for the molecular datasets. And, 40* 1 with ReLU activation in our experiments for the Sinusoids. However these are part of the network architecture and as such must be selected by the experimenter. We have clarify these details in the experimental updated paper.
>
> Q3  -- Is there an ablation to show the contribution of using learnable kernel network vs hand-picked kernel for ADKL-GP and ADKL-KRR?
> R3 - Unfortunately there is no such experiments in the current version of the paper. However, during our experiment journey that , we tried to hand picked the kernel for ADKL-GP and ADKL-KRR and we did not have any success doing so.
>
> Q4  -- (line 181) "where U is a set of unlabeled inputs" Where does the unlabeled inputs come from?
> R4 - No, U is not a set of unlabeled inputs rather a set of pseudo-representations that we learned during the outer-loop of meta-learning. We are responsible for this confusing in our attempt to explain why U is important and where is comes from the version of the paper that you reviewed. But, now we have updated  the manuscript to make this clearer.
>
> Q5  -- Is there an ablation about |U|  <= 50 and its effect?
> R5 - Yes, we tried |U| = 0, 20 and 50 in our experiments and show the relative performance between of |U| = 20 and 50 compared to  |U| = 0 in Figure 4 - b and d. However, we did not extend beyond 50 as this increases the computation time and we could not afford this cost for our experiments due to resource constraints.
>
> Q5- The ProtoMAML code provided by the authors has only a few lines of code. And in the comment, it says: "It turns out that ProtoMAML is the same as the light version of BMAML with one particle." Such implementation detail should be mentioned in the main paper.
> R5 - Thank you for bringing our attention to some of the more confusing details of our implementation. ProtoMAML was originally developed for classification. We modified it slightly for regression purposes by modifying the prototypical layer with a linear layer whose initialisation is learned during the outer training loop and adapted using gradient descent within the inner loop. As such, the name ProtoMAML might be somewhat improper (but we employed it in a natural way to generalise the algorithm to regression).
>
> The correspondence between that particular variant of ProtoMAML and BMAML comes from the following fact: the memory-efficient version of BMAML is as a two-part network with a feature extractor, which is shared between tasks and updated in the outer loop of the meta-learning procedure and a final layer, which is updated during the fast-adaptation (inner loop) using an initialisation learned during the outer loop. When using multiple particles, there are multiple versions of the final layer that are learned and used to perform predictions but, when only one particle is used, this corresponds to the aforementioned ProtoMAML.
>
> However, having said this, we have removed this algorithm from the updated manuscript to avoid confusing the reader and also to avoid offending the authors of ProtoMAML with our overly simplistic generalization of their method to regression.

---

> ### Author Response · Authors · 2019-11-15
> **response to #5 --part 2**
>
>
> Q6 - How each meta-train episode is formed? Did they stochasticall sample a subset of molecules for each episode?
> R6 - As mentioned in the updated version of our manuscript, the episodes are formed as follows :
> Before training, tasks are divided between meta-train and meta-test. Once that separation is done it will not change. Note that the tasks are divided the same way across the experiments (depending only on the seed) to ensure comparable results.
> At train time, each training batch consists of a fixed number of randomly sampled tasks from the meta-training set. For each task, we build the episode by defining the support and the query sets, whose examples are randomly sampled (without replacement) from the original task dataset.
>
> This sampling scheme means that any one task cannot be seen both in the meta-train and the meta-test sets. However, the same training example can be used in the support and the query set, although not during the same batch.
>
> Q7 - What is the total number of tasks in the meta-train and meta-test split respectively? Are there any overlapping between the two splits?
> R7 - We used 1000 tasks for testing for the molecular benchmarks and 1250 for the Sinusoids datasets. The remaining tasks for each collection are used for meta-training and validation. There is no overlap between meta-train, meta-valid, and meta-test.
>
> Q8 - Section 4 (line 201) "DKL methods lie between neural networks and kernel methods". To me, at high-level, DKL adds kernel learning as a differentiable layer in the end of a neural network. See how a similar model is categorized in Bertinetto et al. ICLR 2019 paper "Meta-learning with differentiable closed-form solvers":  "The main idea is to teach a deep network to use standard machine learning tools, such as ridge regression,as part of its own internal model, enabling it to quickly adapt to novel data." The delta of the paper is adding 1) Gaussian Process and 2) task-specific adaptation. If Bertinetto et al. categorize their method as part of a deep network then I think so should this work.
> R8 - We did not mean that DKL methods were outside of the scope of deep neural networks but rather that they incorporated ideas from both neural networks as well as kernel methods. The method itself is indeed part of the neural network family.
>
> Q9 - Section 5 (line 272) "Fig 2 highlights three aspects of the collections that make them better benchmarks for evaluating the readiness of FSR methods for real-world applications relative to toy collections." What is the other benchmarks the authors are comparing with? Are they generic benchmarks for all types few-shot regression tasks/applications beyond biological assays (e.g. computer vision tasks like: object detection). Maybe defining the proposed benchmark as "complimentary" and "bio-assay application focused" would be more appropriate.
> R9 - Our intent with this statement was simply to emphasize that most existing benchmarks (1D or 2D function regression tasks or image-related tasks) do not have the characteristics intrinsic to bioassay modelling tasks that we believe are shared by other real-world applications of few-shot regression (e.g. hyperparameter optimization, biomedical survival analysis, etc). However, you are correct in pointing out that the benchmarks we propose herein are complementary and bio-assay application focused (rather than better than existing benchmarks) and have modified the main text to reflect this.
>
> Thank you for the additional feedback on the presentation. We have updated the manuscript to improve its clarity on all aspects referenced in your comments and hope that any confusion has been removed.

---

### Official Review · AnonReviewer2 · 2019-11-01
**Official Blind Review #2**

**Rating:** 3

**Review:**

This is an interesting paper that proposes the use of few shot regression to predict complicated experimental measurements such as protein-ligand binding affinity from very small, noisy real-world datasets. It is great that the authors make the effort to apply their approach to these important questions in drug-discovery. However, the paper as currently written is difficult to follow, and in particular it is hard to distinguish between places where existing methods are combined from novel contributions made by this paper. It would be helpful if the authors could explicitly delineate the novel contributions that they make. Moreover, there is a large body of work in the drug discovery literature that uses sparse experimental data on the interactions of multiple target proteins and multiple ligands to build models that predict the outcome of biological assays for held out protein targets, where this problem is known as drug-target interaction prediction, but these papers are not referenced in this work (e.g. reviewed in Chen et al. Molecules 23(9):1-15, 2018, Ezzat et al. 2017, 2018, 2019).

In addition, it is hard to understand the results of the experiments that the authors carry out in this paper using data from BindingDB and PubChem. I don't understand the scaling that is applied to the MSE metric for the binding or antibacterial datasets. Why are the reported MSEs so low for all the methods? What does this metric mean? If the targets are first log2-scaled, then scaled linearly, then how different are they after this process? Given the tiny amount of data available, it seems surprising to me that the MSE is so low. From the brief description given, these problems appear significantly harder than the artificial Sinusoids task, yet the reported MSEs are orders of magnitude smaller. How is molecule similarity measured in Figure 2c? It would be useful for the authors to visualize performance for the Binding and Antibacterial tasks - for example how different are the different protein targets? What are the molecular ligand structures in the train and test set in each case - could the authors provide some examples? Using random splits into train and test likely means that some test data points are very close to train data points - can the authors stratify their analysis to provide some insight into the performance beyond the average MSE?

Overall this paper makes a potentially interesting contribution, but it is not well situated within the drug discovery literature, and the results are not explained in enough detail to be understandable by experts in the drug discovery field.


**Experience Assessment:**

I have published one or two papers in this area.

**Review Assessment: Checking Correctness Of Derivations And Theory:**

I assessed the sensibility of the derivations and theory.

**Review Assessment: Checking Correctness Of Experiments:**

I carefully checked the experiments.

**Review Assessment: Thoroughness In Paper Reading:**

I read the paper at least twice and used my best judgement in assessing the paper.

---

> ### Author Response · Authors · 2019-11-15
> **response to #2 --part 1**
>
> First, we would like to say that we appreciate your thorough review, and all comments and questions regarding our manuscript. In the rest of this response, we try to address all questions and comments.
>
> Q1 - However, the paper as currently written is difficult to follow, and in particular it is hard to distinguish between places where existing methods are combined from novel contributions made by this paper. It would be helpful if the authors could explicitly delineate the novel contributions that they make.
> R1 - There were indeed certain sections that required polishing and we have now updated our manuscript to make the methodological contribution easier to follow and understand. We sincerely hope that the confusion has been alleviated, but we would nonetheless like to briefly clarify our technical contribution and its novelty. We begin the paper by observing that deep kernel learning is a simple yet effective method to learn from few samples. However, given that using the same hand-selected kernel for all tasks has certain limitations, we proposed ADKL to obtain the suitable kernel during inference. To do so, we learn task representations by maximizing an estimate of the mutual information between the support and query sets of a task (doing so is completely novel and ablation studies show that is critical to the model performances). We then use these representations to infer the adequate kernels for each task. This process involves learning a neural network and leveraging some pseudo-inputs that lie in the feature space on which this network operates (this also all novel). Finally, the proposed algorithm was applied to bioassay modelling, representing (to the best of our knowledge) the first application of few-shot learning methods to this domain, with results indicating that our approach is better than existing methods for such tasks.
>
> Q2 - Moreover, there is a large body of work in the drug discovery literature that uses sparse experimental data on the interactions of multiple target proteins and multiple ligands to build models that predict the outcome of biological assays for held out protein targets, where this problem is known as drug-target interaction prediction, but these papers are not referenced in this work (e.g. reviewed in Chen et al. Molecules 23(9):1-15, 2018, Ezzat et al. 2017, 2018, 2019).
> R2 - Our experimentations with BindingDB can indeed be viewed as an instance of a drug-target interaction (DTI) prediction problem. However, there are some noticeable differences between our application of meta-learning to BindingDB and the existing literature in DTI. First, in a classical DTI problem, the target protein is known and is used as part of the model construction. ADKL does not use this information and relies solely on the training set of molecules to build a model. Second, this work extends beyond the scope of DTI predictions given that the use of meta-learning allows us to deal with additional types of tasks. For example, our experiments with antibacterial assays involve only phenotypic assays (i.e. no specific protein target is known). Finally, this work focuses on learning when data is particularly scarce and, as such, we have developed a model can be applied across any bioassay modelling task for which data is limited. This is in contrast to many DTI methods, which tend to be used only to build screening tools to discover new potential ligands for a given protein or new targets for existing molecules.
>
> Q3 - In addition, it is hard to understand the results of the experiments that the authors carry out in this paper using data from BindingDB and PubChem. I don't understand the scaling that is applied to the MSE metric for the binding or antibacterial datasets. Why are the reported MSEs so low for all the methods? What does this metric mean? If the targets are first log2-scaled, then scaled linearly, then how different are they after this process?
> R3 - We first wish to clarify how the MSE is computed for all Tables 1-3: each task is partitioned into query and support sets, then the support set is used to generate a model which is evaluated on the query set to compute the MSE. This process is repeated 30 times per task and the average over the repetitions per task and over all tasks is the value shown in Tables 1-3.
> For BindingDB and Antibacterial, the MSE metric is computed using the normalised version of the targets (first log2-scaled and then mapped to the [0-1] interval). This normalisation is done because the original target scale for each task goes from 10^-3 to 10^6. It is required to avoid the meta-training being biased towards a specific group of tasks which will have a larger MSE because of the scale of their targets. The values are low and expected to be so due to the scaling but, as we understand that this may lead to some confusion, we now report the RMSE for Antibacterial and BindingDB as well as for the toy dataset.

---

> ### Author Response · Authors · 2019-11-15
> **response to #2 -- part 2**
>
>
> Q4 - Given the tiny amount of data available, it seems surprising to me that the MSE is so low. From the brief description given, these problems appear significantly harder than the artificial Sinusoids task, yet the reported MSEs are orders of magnitude smaller.
> R4 - It can indeed be surprising that the MSE for molecular tasks is lower than the MSE for the Sinusoids tasks. However, as the MSE is scale dependant, it is expected to be so because the targets of the molecular tasks are pre-processed and rescaled to the [0-1] interval, but those for Sinusoids are from [0.1, 5] (range of amplitude). We did not preprocess the Sinusoids as this is the setup used by all previous papers.
>
> Q5 - How is molecule similarity measured in Figure 2c?
> R5 - The similarity measure used in Figure 2C is the Tanimoto similarity computed on the molecular fingerprint of the molecules (ECFP4, folded back to 4096 bits). (David Rogers and Mathew Hahn, 2010)
>
> Q6 - It would be useful for the authors to visualize performance for the Binding and Antibacterial tasks - for example, how different are the different protein targets? What are the molecular ligand structures in the train and test set in each case - could the authors provide some examples?
> R6 - As the intended audience for this paper is primarily the few-shot learning community, we feel that such a detailed analysis would make the paper harder to digest for most readers. However, as all the datasets and code to reproduce all our experiments will be publicly available, we invite any interested reader to carry out such experiments at their own accord.
>
> Q7 - Using random splits into train and test likely means that some test data points are very close to train data points - can the authors stratify their analysis to provide some insight into the performance beyond the average MSE?
> R7 - Indeed, randomly splitting each task into train and test could lead to some test data points being very close to other train data points. However, as shown by Figure 2 (left panel), most data points for any given bioassay are comparatively more similar than data points from different bioassays. This means that trying to stratify the internal train/test split for most tasks will not avoid the problem that you have pointed out. However, we understand that showing the performance beyond the average MSE will be insightful and have therefore provided the results of pairwise statistical analysis that compare ADKL to all other algorithms.
>
> Overall this paper makes a potentially interesting contribution, but it is not well situated within the drug discovery literature, and the results are not explained in enough detail to be understandable by experts in the drug discovery field.
> We appreciate that the manuscript, in its current form, is not detailed enough to be understood by experts in drug discovery. However, as this paper was submitted to ICLR, we expect most readers to come from the few-shot learning and computational chemistry communities, and we have therefore intentionally limited drug discovery related details (and associated jargon) so the intended audience may more easily follow. Moreover, as we wish to bring the attention of the few-shot learning community to the applications available in drug discovery, as well as present a new few-shot modelling technique, we feel that this level of detail might discourage the community from taking further interest in these types of applications.

---

### Official Review · AnonReviewer6 · 2019-11-02
**Official Blind Review #6**

**Rating:** 8

**Review:**

The submission is at the intersection of few-shot learning, kernel regression methods, and computational biology.
The main contributions are:
  - A few-shot learning algorithm combining several ideas and features:
    - Combining metric learning (shared across tasks) and kernel regression within each task
    - Learning a task representation by maximizing an estimate of the mutual information between the train (support) and valid (query) sets of a task
    - The addition of learned, synthetic "pseudo-examples"
  - Two datasets for few-shot regression, from real-life biological assays
The proposed algorithm outperforms (or is competitive with) mainstream few-shot learning methods on a synthetic 1D regression dataset, as well as the two proposed datasets.

This paper should be accepted, because it significantly expands the field of few-shot learning by proposing both a novel problem to tackle (few-shot regression from a high-dimensional, noisy input), with public datasets, and novel algorithm to solve it (combining several recent advances in different sub-fields of machine learning).

Overview
The overall problem is clearly stated, as well as its main challenges: noisiness of the data, different behaviors of the same input across tasks.
Despite the complexity of the proposed algorithm, its different pieces and their motivations are clearly motivated, introduced, and tested in ablation studies. I liked the clarity of section 2.
The "related work" section is clear and presents a good overall picture of the field. Additional papers that may be of interest:
  - Learned hallucination (Low-shot learning from imaginary data, Wang et al., CVPR 2018) seems to relate to the "pseudo-representations"
  - TADAM (Task dependent adaptive metric for improved few-shot learning, Oreshkin et al., NeurIPS 2018) is another example of task representation used in conjunction with metric learning (in the context of classification, not regression, though)
The proposed datasets are an interesting new benchmarking task, that naturally requires learning a task description, I hope it will get traction.

Questions:
  - The backpropagation through the kernel regressor (in order to train the parameters of the embedding function and other networks) seems unexpectedly straightforward. Is that only because there is a closed-form solution for the optimal regressor? Were the specific algorithms (KRR, GP) chosen because of that? Or could something like MetaOptNet (Meta-Learning with differentiable convex optimization, Lee et al., CVPR 2019) be used to relax that constraint?
  - What "generalization guarantees of kernel-based models" (l. 122) would be applicable here? Do they hold even when the kernel is applied on top of a learned embedding ($\phi_\theta$), or even when it depends on a trained model itself ($C_t$)?
  - What does "correlations > 0.8" mean exactly? (l. 258, 263)
  - In the "Binding" dataset, is it possible that the same protein is used in different bio-assays? In that case, are the different experiments "merged" into the same task, or be considered different tasks? Could it be possible that tasks involving the same protein would be in different (meta-)splits, or has that been taken care of during the data collection?
  - In the meta-test splits, how are examples split between the train/support and valid/query parts of each task?
  - How were the specific architectures of the different neural networks designed, or selected? Between $\phi$, v, r, $MLP_\rho$, the space of hyper-parameters seems huge, and the effects of these choices might be drastic.

Additional feedback
  - The caption of Figure 1 could (re-)introduce a definition of the notation. U and C_t for instance have not been introduced yet.
  - I'm not sure I agree that FSDKL only "share[s] characteristics with the metric learning framework", I see it more as being in that framework, but incorporating other elements as well (like other methods do, e.g., TADAM or RelationNet).
  - The horizontal axis of Figure 4 (a) and (c) are not clear until we see Table 4 of the appendix, and suggest an ordering of the different configurations, rather than 10 different categories. I'm not sure how to improve it though, maybe letters instead of numbers?

On notation:
  - On l. 155, is $\phi$ the same as $\phi_\theta$, or a different embedding function for x?
  - In Eq. 11, should $D^t_{trn}$ be $D^{t_j}_{trn}$ instead in both terms? Similarly, $D^{t_j}_{val}$ in the first term?
  - In Eq. 12, and l. 173, $\phi_x'$ suggests it is a different embedding $\phi'$ of the same $x$, if we want to convey that there are two inputs x and x' instead, $\phi_{x'}$ may be clearer.
  - In Fig. 4 (a) and 5, the vertical axis is labeled $\gamma_{mine}$ instead of (I assume) $\gamma_{task}$.

**Experience Assessment:**

I have read many papers in this area.

**Review Assessment: Checking Correctness Of Derivations And Theory:**

I assessed the sensibility of the derivations and theory.

**Review Assessment: Checking Correctness Of Experiments:**

I assessed the sensibility of the experiments.

**Review Assessment: Thoroughness In Paper Reading:**

I read the paper thoroughly.

---

> ### Author Response · Authors · 2019-11-15
> **Response to #6**
>
> First, we would like to thank you for your thorough review, your kind words regarding our manuscript, and the additional related work that you have pointed us towards. In the remainder of our response, we try to address all questions and comments.
>
> Q1  - The backpropagation through the kernel regressor (in order to train the parameters of the embedding function and other networks) seems unexpectedly straightforward. Is that only because there is a closed-form solution for the optimal regressor?
> Were the specific algorithms (KRR, GP) chosen because of that? Or could something like MetaOptNet (Meta-Learning with differentiable convex optimization, Lee et al., CVPR 2019) be used to relax that constraint?
> R1 - Yes, the straightforwardness of the backpropagation of the kernel regressors in ADKL is only due to the fact that we choose algorithms that admit closed form solutions (GP and KRR). The backpropagation is easily done in such cases using the chain-rule and the automatic differentiation algorithms that exist in popular frameworks such as Pytorch or Tensorflow. However, an algorithm that does not admit a closed form solution could be used as was the case in MetaOptNet and also in (Meta-learning with differentiable closed-form solvers, ICLR2019).
>
> Q2  - What "generalization guarantees of kernel-based models" (l. 122) would be applicable here? Do they hold even when the kernel is applied on top of a learned embedding (), or even when it depends on a trained model itself ()?
> R2 - When the meta-training is over, using ADKL for get a  model is the same as using any other kernel function designed by hand or using a more sophisticated procedure and then perform  KRR. Thus, it is our view that Corollary 13.10 of [1]  is still applicable even if the kernel method is applied on top of a learned embedding function. It is worth recalling that this corollary bounds the true risk of a model obtained by minimizing Equation 1 (see paper) and can be applied to both KRR and GP.
>
> [1] Shalev-Shwartz, Shai, and Shai Ben-David. Understanding machine learning: From theory to algorithms. Cambridge university press, 2014.
>
> Q3  - What does "correlations > 0.8" mean exactly? (l. 258, 263)
> R3  - The correlation threshold was used to remove tasks that were too similar in order to avoid information leakage between training and testing. This is now clarified in the manuscript and we provide a summary of the process below.  For any tasks pair in a given collection that share a high proportion of molecules and has highly correlated targets, as measured by the Pearson correlation coefficient (PCC), we remove one of them. We used  a threshold of  0.8 to indicate a high correlation between tasks.
>
> Q4  - In the "Binding" dataset, is it possible that the same protein is used in different bio-assays? In that case, are the different experiments "merged" into the same task, or be considered different tasks? Could it be possible that tasks involving the same protein would be in different (meta-)splits, or has that been taken care of during the data collection?
> R4  - The same protein can indeed be present in different bio-assays. However, different assays featuring the same protein often involve different molecules, experimental setups or measures. Therefore, we did not perform any form of pre-processing to avoid having the same protein involved in the meta-training and the meta-testing set multiple times. Also, by filtering the highly correlated tasks as described above, we can ensure that if the same protein is involved in different tasks, they are sufficiently different so as not to bias the performance being measured in any way.
>
> Q5  - In the meta-test splits, how are examples split between the train/support and valid/query parts of each task?
> R5 - The meta-test set is randomly split by sampling K=5, 10, 20 examples for the train partition and using the remaining for the test partition. To obtain the performance of an algorithm for each task, we repeat this train/test split 30 times and report the average over all the splits. The results in Tables 1-3 are the average over all the tasks using this procedure.
>
> Q6  - How were the specific architectures of the different neural networks designed, or selected? Between , v, r, , the space of hyper-parameters seems huge, and the effects of these choices might be drastic.
> R6 - The choice of hyper-parameters is indeed a topic in itself. Herein, we first performed the ablation studies on the toy datasets to better understand the effects of each hyperparameter on the learning process and performance. The results obtained gave us some insight into the values of  hyper-parameters to try for the real world datasets.  The results were then obtained after a light hyper-parameter search (which was also carried out for other baseline algorithms to avoid biasing results in favour of our method).
>
> Thank you for the additional feedback on the figures and notation. We have updated the manuscript  consequently

---

### Official Review · AnonReviewer4 · 2019-11-04
**Official Blind Review #4**

**Rating:** 6

**Review:**

This paper presents a new framework for solving few-shot regression problems. The proposed framework is based on deep kernel learning, which lies in the intersection of neural networks and kernel methods. The authors introduced adaptive deep kernel learning, which learns kernel for multiple task collection and computes the correct kernel for a task in a data-driven manner. The method is evaluated on three tasks collections — Sinusoids (synthetic dataset), Binding (real dataset of 5717 task where each task represents a binding affinity of small molecules against a given protein) and Antibacterial (real dataset of 3255 tasks where each task represents antimicrobial activity against given bacterium).


Pros:
1. The proposed framework introduces a new adaptive method for few-shot drug discovery regression problems.
2. The paper addressed the question of uncertainty estimation for drug discovery tasks.


Cons:
1. The authors only evaluate the performance of the proposed method against other DKL-based methods. They do not consider a comparison with widely used in computational chemistry classical machine learning methods such as random forest, gradient boosting, SVM (which is also a kernel method), etc.
2. The experimental results in Tables 1-3 show only marginal improvement for real datasets. Also, it’s not clear if the numbers in the Tables 1-3 are on the original scale or on the transformed scale. To estimate how well the models perform it’s useful to transform the targets back to the original scale.
3. Overall the paper is written in a somehow confusing manner and some details in the description of the experiments important for understanding are omitted.

Questions:
1. How well the proposed method performs against classical machine learning methods?
2. How was MSE in Tables 1-3 calculated?
3. It would be useful to see the histogram of MSEs instead of just a single number.


**Experience Assessment:**

I have published one or two papers in this area.

**Review Assessment: Checking Correctness Of Derivations And Theory:**

I assessed the sensibility of the derivations and theory.

**Review Assessment: Checking Correctness Of Experiments:**

I assessed the sensibility of the experiments.

**Review Assessment: Thoroughness In Paper Reading:**

I read the paper at least twice and used my best judgement in assessing the paper.

---

> ### Author Response · Authors · 2019-11-15
> **respons to #4**
>
> First, thank you for your thorough review and suggestions to improve the paper. In the remainder of our response, we try to address your main questions and comments.
>
> Q1: How well the proposed method performs against classical machine learning methods?
>
> R1: This is a great question as the ultimate goal is to improve not only upon meta-learning methods but also classical methods. To this end, we performed new experiments using two techniques considered to be state-of-the-art in chemoinformatics (1) the Random Forest algorithm with ECFP4 (Extended Connectivity FingerPrints of diameter 4) as molecular input representation [1], and (2) ECFP4 with kernel ridge regression and tanimoto similarity as a kernel function. Overall, new results in the updated Tables 2 and 3 show that ADKL performs better than classical methods on the Antibacterial collection and is competitive on the Binding collection. However, it is worth noting that other meta-learning technique fall significantly behind relative to these strong baselines. These results indicate that although ADKL is significatively superior to other meta-learning methods (as shown by the relative p-value to ADKL-KRR for pairwise comparison at the task level), there remains much room to develop improved meta-learning algorithms which are undoubtedly superior to classical methods in computational chemistry. Indeed, our hope in offering the proposed datasets is that the community can progress towards developing better meta-learning algorithms for regression problems.
>
> Q2: How was MSE in Tables 1-3 calculated? The experimental results in Tables 1-3 show only marginal improvement for real datasets. Also, it’s not clear if the numbers in the Tables 1-3 are on the original scale or on the transformed scale.
>
> R2: During meta-test, each task is partitioned into query and support sets, then the support set is used to generate a model which is evaluated on the query set to compute the MSE. This process is repeated 30 times per task and the average over the repetitions per task and over all tasks is the value shown in Tables 1-3.
> For Table 1 (Sinusoids), the MSE is computed on the original target scale but for Table 2-3 (molecular datasets), we used the transformed scale (doing so it necessary to avoid  the averaging being dominated by tasks whose output domains are large: the original output domain goes from 10^-3 to 10^6).
>
> Q3: It would be useful to see the histogram of MSEs instead of just a single number.
> R3: Indeed, a single value does not give the whole picture when comparing meta-learning algorithms. However, for space limitation, we could not show the histograms in the main text (See appendix 6 please). In the updated version of the paper, we also provide a more detailed overview of the performance of each method (confidence intervals of the MSE after multiple runs, and p-values of pairwise statistical analyses at the task level for every method compared to ADKL-KRR), and we hope that is helpful in your analysis of the results.
>
> [1] Olier, Ivan, et al. "Meta-QSAR: a large-scale application of meta-learning to drug design and discovery." Machine Learning 107.1 (2018): 285-311.

---

### Comment · AnonReviewer1 · 2019-10-22
**I don't quite follow this line of work.**

I unfortunately can't provide technical substance. But I do feel the presentation makes understanding unusually hard.

---

> ### Comment · AnonReviewer1 · 2019-10-28
> **Can't assess this line of work**
>
> I unfortunately can't provide technical substance. But I do feel the presentation makes understanding unusually hard.

---

> > ### Author Response · Authors · 2019-10-28
> > **Thanks**
> >
> > Thank you for your comments. We are happy to try to clarify specific thoughts or ideas as needed.

---

### Author Response · Authors · 2019-11-15
**To all reviewers**

We offer a sincere thank you to all reviewers for their insightful comments and suggestions to improve our paper. We have addressed the primary concerns of each reviewer by responding directly to your comments and have updated the manuscript accordingly. We have also taken special care to clarify any confusing sections in the paper to help with readability. With this global comment, we wish to summarize all major changes to facilitate your ongoing review process.

1 - We have rewritten Section 3 of the paper after it was suggested that it was hard to follow. We hope that the changes will improve understanding of the methodological contribution of our work.

2 - Second, we redid experiments related the molecular benchmarks to provide confidence intervals for the MSE. During these experiments, we changed the number of tasks in the meta-test from 500 to 1000 for each collection to have a better estimate of the meta-test performance of all methods. (Unfortunately, due to resource constraints, the results of ADKL-GP for the Binding collection will be missing for the moment. Thanks for understanding).

3 - We also included new models in the comparison process for these datasets. These methods are considered to be state-of-the-art in chemoinformatics and comparing against them is very insightful regarding how current meta-learning methods perform relative to existing methods for bioassay modelling.

4 - We have added p-values of pairwise statistical tests that compared each algorithm to ADKL-GP and ADKL-KRR. More precisely, for any two algorithms, we did the Wilcoxon ranked test which compared the results of both algorithms for each task to determine if one is significantly better than the other. These p-values are provided in Tables 4-5.

5 - Following the previous points, the analysis of the results has been slightly modified as well.

---

### Decision · Program_Chairs · 2019-12-19

**Decision:**

Reject

**Comment:**

This work applies deep kernel learning to the problem of few shot regression for modeling biological assays. To deal with sparse data on new tasks, the authors propose to adapt the learned kernel to each task. Reviews were mixed about the method and experiments, some reviewers were satisfied with the author rebuttal while others did not support acceptance during the discussion period. Some reviewers ultimately felt that the experimental results were too weak to warrant publication. On the binding task the method is comparable with simpler baselines, and some felt that the gains on antibacterial were unconvincing.
Other reviewers felt that there remained simpler baselines to compare with, for example ablating the affects of learning the kernel with simple hand picking one. While authors commented they tried this, there were no details given on the results or what exactly they tried.

Based on the reviewer discussion, the work feels too preliminary in its current form to warrant publication in ICLR. However, given that there are clearly some interesting ideas proposed in this work, I recommend resubmitting with stronger experimental evidence that the method helps over baselines.